# KV Cache is 1 Bit Per Channel: Efficient Large Language Model Inference with Coupled Quantization

**Tianyi Zhang**
Dept. of Computer Science, Rice University
xMAD.ai
Houston, TX
tz21@rice.edu

**Jonah Yi**
Dept. of Computer Science, Rice University
xMAD.ai
Houston, TX
jwy4@rice.edu

**Zhaozhuo Xu**
Dept. of Computer Science,
Stevens Institute of Technology
xMAD.ai
Hoboken, NJ
zxu79@stevens.edu

**Anshumali Shrivastava**
Dept. of Computer Science, Rice University
Ken Kennedy Institute
ThirdAI Corp.
xMAD.ai
Houston, TX
anshumali@rice.edu

## Abstract

Efficient deployment of Large Language Models (LLMs) requires batching multiple requests together to improve throughput. As batch size, context length, or model size increases, the size of key and value (KV) cache quickly becomes the main contributor to GPU memory usage and the bottleneck of inference latency and throughput. Quantization has emerged as an effective technique for KV cache compression, but existing methods still fail at very low bit widths. Currently, KV cache quantization is performed per-channel or per-token independently. Our analysis shows that distinct channels of a key/value activation embedding are highly interdependent, and the joint entropy of multiple channels grows at a slower rate than the sum of their marginal entropy, which implies that per-channel independent quantization is sub-optimal. To mitigate this sub-optimality, we propose Coupled Quantization (CQ), which couples multiple key/value channels together for quantization to exploit their interdependence and encode the activations in a more information-efficient manner. Extensive experiments reveal that CQ compares favorably with existing baselines in preserving model quality, and improves inference throughput by 1.4–3.5× relative to the uncompressed baseline. Furthermore, we demonstrate that CQ can preserve model quality reasonably with KV cache quantized down to 1 bit.

## 1 Introduction

Large Language Models (LLMs) have showcased remarkable generalization abilities across various tasks without needing specific fine-tuning [30]. These impressive capabilities have empowered LLMs to find applications in numerous domains [19]. However, the high computational demands and prohibitive deployment costs of LLMs create significant barriers, hindering their widespread adoption [19, 4]. Particularly, as LLMs move towards larger model size [14] and longer context length [40], they require faster hardware accelerators such as graphics processing units (GPUs) with higher memory capacity for efficient inference. Hence it is crucial to develop approaches for reducing the computational costs and memory requirement of LLMs.

38th Conference on Neural Information Processing Systems (NeurIPS 2024).

Key and value (KV) caching [47] has proven to be an effective technique for accelerating LLM inference without affecting model quality. In autoregressive LLMs, KV caching works through trading off memory to save computations: the key and value activations of all previous tokens in the current sequence are cached in memory to avoid their recomputation for generating the next token. However, KV cache can quickly overwhelm the memory capacity of GPUs as context length or batch size increases, since its storage scales linearly with these two factors. Consider the OPT-175b model [43], storing its KV cache for 128 sequences of 2048 tokens requires 1.2 terabytes of memory, which is around $3.5\times$ the storage of its weights. Because inference throughput scales with batch size, the substantial memory demands of KV caching can become a significant bottleneck to throughput. In addition, as KV cache is not shared across sequences within a batch, it has a low compute-to-memory ratio, making reading the KV cache from GPU memory the primary source of latency as opposed to the attention computation [15]. KV cache compression can bring the following benefits: 1. speeding up LLM inference by improving the compute-to-memory ratio, 2. improving the serving throughput by fitting more sequences into memory and hence enabling larger batch sizes, 3. lowering the GPU requirements for inference for a given batch size, context length, and model size. Existing approaches typically achieve compression of KV cache through token eviction [47, 24] or activation quantization [25, 15]. While these methods can preserve model quality at moderate compression rates ($4\times$ compression or 4 bits per activation), model quality quickly deteriorates at high compression rates ($16\times$ compression or 1 bit per activation). In this work, we leverage the interdependency between key/value channels, an insight overlooked by existing approaches, to achieve higher compression rates of KV cache while maintaining model quality.

Our approach is motivated by the observation that distinct channels within the same key/value activation embedding are highly interdependent and correlated (see our analysis in Section 3.1). It is hence more information-efficient to encode multiple channels of KV cache at once, which we call *channel coupling*. Existing solutions, in contrast, only employ per-channel or per-token quantization strategies [25, 15] for compressing KV cache, which is not optimal for exploiting the dependency between channels. As a result, we observe significant model quality degradation at the extreme compression rate of 1 bit per activation. By leveraging channel coupling, we enable compression at the level of 1-bit quantization of KV cache while preserving model quality. In Figure 1, we show the perplexity of two models from the LLaMA family [36, 37] on WikiText-2 [27] under 1-bit quantization with varying numbers of

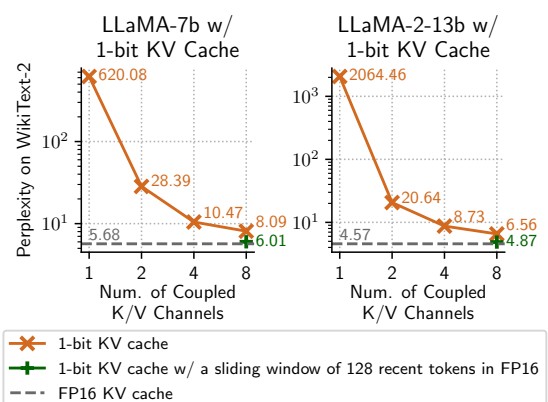

Figure 1: Perplexity of LLMs with 1-bit quantized KV cache approaches the uncompressed FP16 performance as the number of coupled K/V channels increases.

coupled channels. The full experimental setup is presented in Section 4. Coupling more channels significantly enhances model quality, as the perplexity quickly approaches the performance using uncompressed FP16 KV cache. By further combining KV cache quantization with a sliding window of 128 recent tokens cached in full precision, we achieve a negligible 0.3–0.33 increase in perplexity with 1-bit KV cache.

We summarize our contributions as follows.

1. We observe the phenomenon that distinct channels within the same key/value activation embedding share a high amount of dependency or mutual information, which has not been leveraged by existing approaches.

2. We propose Coupled Quantization (CQ), a novel KV cache quantization method that jointly encodes multiple key/value channels to exploit the dependency across channels.

3. Through extensive experiments, we demonstrate the effectiveness of CQ at preserving model quality and speeding up LLM inference against competitive baselines. Furthermore, we demonstrate CQ reasonably preserves model quality at an extreme level of 1-bit quantization.

## 2 Background

This section introduces the relevant background information including the KV caching technique and per-channel quantization.

### 2.1 LLM Attention and KV Cache

Decoder-only transformer-based LLMs employ masked self-attention [38], in which activations of the current token only depend on the previous tokens and are unaffected by future ones. This property enables training parallelism for the next-token prediction objective, and gives rise to the KV caching technique for efficient decoding during inference. Consider the decoding step for the $t$-th token in a single head of attention in an LLM. The input embedding of the $t$-th token (a column vector), $e_t$, goes through three distinct transformations to become key, query, and value activation embeddings $f_K(e_t), f_Q(e_t), f_V(e_t)$, where the transformations $f_K, f_Q, f_V$ are composed of linear projection and positional encoding methods such as RoPE [35]. The output embedding of attention for the $t$-th token is computed as

$$\text{attention}(e_t) = \left[\begin{array}{ccc} f_V(e_1) & \dots & f_V(e_t) \end{array}\right] \text{softmax}\left(\left[\begin{array}{ccc} f_K(e_1) & \dots & f_K(e_t) \end{array}\right]^\top f_Q(e_t) \Big/ \sqrt{d}\right) \quad (1)$$

where $d$ is the dimensionality of $f_K(e_t)$. Computing the output embedding of the current token requires the key and value activation embeddings of all previous tokens, $f_K(e_i)$ and $f_V(e_i)$ where $i \in \{1, \dots, t-1\}$. These embeddings are cached in memory from previous decoding steps to eliminate redundant computations, a process known as KV caching. The size of KV cache can be calculated as $b \times n \times l \times 2 \times h \times d$ floating-point numbers, where $b$ is the batch size, $n$ is the number of tokens in each sequence, $l$ is the number of layers in the model, 2 is for key and value, $h$ is the number of key/value attention heads, and $d$ is the dimensionality of a single head of key/value activation embedding. As batch size, context length, or model size increases, the size of the KV cache can quickly overwhelm the limited GPU memory. KV cache bottlenecks inference throughput since it limits the maximum batch size, and it is a major contributor to latency due to the low compute-to-memory ratio [15].

### 2.2 Per-Channel Quantization

Existing KV cache quantization methods [15, 25] employ per-channel quantization for keys and per-token quantization for values. Per-channel and per-token quantization are similar, except the direction along which the quantization centroids are learned (or the direction along which the scaling factor and zero-point are determined for uniform quantization). Keys are quantized per-channel based on the observation that certain key channels have significantly higher magnitudes than others, while values are quantized per-token because value channels have no such outliers. In non-uniform per-channel quantization, a set of centroids is learned for each channel. Suppose $A$ is a key or value activation matrix, and let $A_{i,:}$ denote the $i$-th channel of $A$. Then, non-uniform $b$-bit per-channel quantization aims to learn a set of centroids $C_i^\star \subset \mathbb{R}$ for each channel $i$ of $A$ through the objective

$$C_i^\star = \underset{\substack{C \subset \mathbb{R} \\ |C| = 2^b}}{\arg\min} \left\| A_{i,:} - q(A_{i,:}) \right\|_2^2 \quad (2)$$

where $q$ quantizes each value in $A_{i,:}$ to the nearest centroid in $C$.

## 3 Methodology

In this section, we motivate our proposal using information theory and introduce the Coupled Quantization (CQ) approach for KV cache compression.

### 3.1 Motivations

Our proposed approach is inspired by concepts in information theory [33]. We consider channels in a key/value activation embedding as random variables $X_1, X_2, \dots$. The amount of information (or uncertainty) in channel $X$ can be measured by *entropy*, defined as $H(X) = -\int_{\mathbb{X}} p(x) \log_2 p(x)\, dx$,

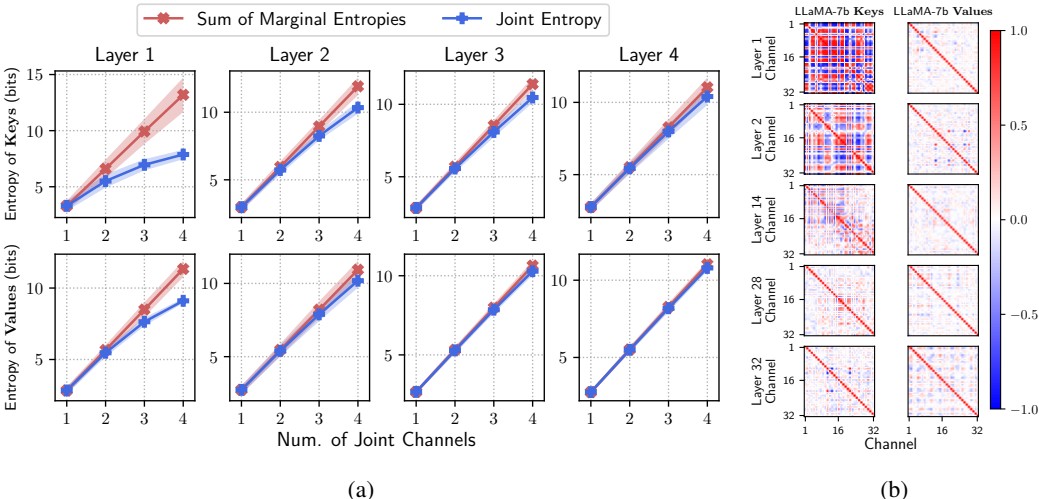

(a)

(b)

Figure 2: (a) Growth rate of joint entropy versus sum of marginal entropies of the key/value activation embeddings of LLaMA-7b on 262k tokens of WikiText-2. Entropy is estimated using Equation 4. The slower growth rate of joint entropy implies that quantizing more channels together is more information-efficient than quantizing fewer channels. (b) Correlation matrices of the first 32 channels of 5 layers of LLaMA-7b key and value activation embeddings on WikiText-2. Channel pairs exhibit high levels of linear dependency, shown by high magnitudes of the correlation coefficients.

where $p(\cdot)$ is the probability density function and $\mathbb{X}$ is the support of $X$. $H(X)$ measures the theoretical number of bits needed for losslessly encoding the channel $X$, hence it can be used to gauge how "quantizable" a channel is: if $H(X_1) < H(X_2)$, then channel $X_1$ may be quantized to fewer bits than channel $X_2$ with the same quantization error.

Our insight is that different channels from the same key/value activation embedding may be interdependent, which would reduce the number of bits required for jointly encoding multiple channels together compared to encoding them independently. The total amount of information (or uncertainty) in two channels $X_1, X_2$ is measured by *joint entropy*, defined as $H(X_1, X_2) = -\int_{\mathbb{X}_1} \int_{\mathbb{X}_2} p(x_1, x_2) \log_2 p(x_1, x_2)\, dx_2\, dx_1$, where $p(\cdot, \cdot)$ is the joint probability density function. Equivalently, the joint entropy of two channels is the difference between the sum of their marginal entropies and their mutual information, i.e., $H(X_1, X_2) = H(X_1) + H(X_2) - I(X_1, X_2)$, where $I(\cdot, \cdot)$ is a non-negative quantity for measuring the mutual dependency of two random variables. Thus, we have

$$H(X_1, X_2) \leq H(X_1) + H(X_2) \tag{3}$$

which implies the number of bits needed for jointly encoding two channels is no more than the total number of bits needed for encoding them independently. Previous works have demonstrated that deep neural networks [16] and attention-based networks [9] tend to produce low-rank embeddings, which suggests that channels of key/value embedding in LLM may exhibit high amount of mutual dependency.

It is hence beneficial to measure the difference between the sum of marginal entropies of multiple key/value channels and their joint entropy. A significant difference would suggest that encoding these channels together is more information-efficient than encoding them independently. However, it is difficult to derive the exact entropy or joint entropy of channels, since their probability density functions are not known. Therefore, we employ the "binning" trick [21] to estimate entropy. We first observe an empirical distribution of key and value channels by saving the KV cache on a dataset, and partition the support of each channel into equally sized bins. Then, values of each channel are discretized to the index of the bin they fall into. Finally, the joint entropy of $n$ channels $X_1, \ldots, X_n$ is estimated with the Riemann sum,

$$H(X_1, \ldots, X_n) \approx \sum_{x_1 \in \mathbb{B}_1} \cdots \sum_{x_n \in \mathbb{B}_n} \hat{p}(x_1, \ldots, x_n) \log_2 \hat{p}(x_1, \ldots, x_n) \tag{4}$$

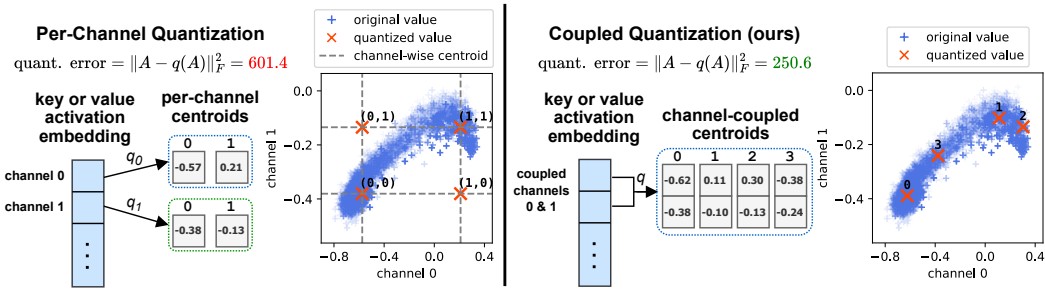

Figure 3: Per-channel quantization (left) and our proposed Coupled Quantization (right). The 1-bit quantization results on the first two channels of the first-layer key activation embeddings of LLaMA-7b on the WikiText-2 dataset are shown. CQ leverages the dependency between channels to achieve lower quantization errors than per-channel quantization.

where $\mathbb{B}_i$ is the support of the binned or discretized $X_i$ and $\hat{p}(\cdot)$ is the empirical probability mass function. Specifically, we divide the channels of key and value embeddings of LLaMA-7b [36] into non-overlapping groups each containing $c$ contiguous channels, where $c \in \{1, 2, 3, 4\}$, and estimate the joint entropy and the sum of marginal entropies of each group. The support of each channel is partitioned into 16 equally sized bins. Figure 2a shows the mean and standard deviation of the estimated joint entropy and sum of marginal entropies of four layers of LLaMA-7b on 262k tokens of the WikiText-2 dataset [27], averaged over groups. We only show a maximum group size of 4, since increasing the group size requires saving exponentially more key and value embeddings to avoid empty bins and maintain estimation quality. As shown in Figure 2a, the sum of marginal entropies grows at a linear rate while the joint entropy increases slower at a sub-linear rate. This implies that as the number of jointly quantized channels increases, the total amount of information needed for encoding them decreases. This phenomenon is the foundation that motivates our proposed approach.

In addition to studying the marginal and joint entropy, we also analyze the linear relationships between channels of a key/value activation embedding using Pearson correlation coefficient. Figure 2b presents the correlation matrices for the first 32 channels of 5 layers of LLaMA-7b keys and values on WikiText-2. The key and value channels exhibit high levels of linear dependency, and are clearly not independently distributed, as shown by high magnitudes of the correlation coefficients. In Section M of the appendix, we include the correlation matrices of all layers of LLaMA-7b, and present scatter plots to visualize the patterns in key and value activations.

## 3.2 Coupled Quantization

Motivated by the finding that distinct key/value channels exhibit high amounts of dependency, we propose Coupled Quantization (CQ), an information-efficient KV cache quantization approach that couples multiple key/value channels for quantization. More concretely, channels of a key or value activation embedding are divided into equally sized, non-overlapping groups of contiguous channels. The channels in each group are *coupled*, as they are jointly quantized and share a single quantization code. For each group of coupled channels, a distinct set of multi-channel centroids are learned, where each centroid has dimensionality equal to the number of channels in that group. When quantizing a key or value activation embedding, each channel group is quantized to the nearest centroid in terms of L2 distance. We use the CQ-<c>cb notation to denote the configuration of channel coupling and quantization bit width, where <c> is the number of channels in each group and  indicates the number of bits in a quantized code for a group. For example, CQ-4c8b means that every 4 contiguous channels are coupled together and each coupled group shares an 8-bit code, which is equivalent to 2-bit per-channel quantization in terms of storage overhead of quantized codes. An illustrative comparison of per-channel quantization and CQ is shown in Figure 3. Although previous works [15, 25] opt to quantize keys per-channel and values per-token, we adopt channel-coupled quantization for both keys and values, which we empirically show is effective for both in Section 4.3. CQ quantizes keys before the positional encoding such as RoPE [35] is applied, which increases the quantization difficulty by introducing more outliers in key activations [15, 25].

### 3.2.1 Centroid Learning

In CQ, the multi-channel centroids for each channel group are learned offline on a calibration dataset by leveraging uniform clustering or second-order-information-informed clustering. Specifically, for uniform centroid learning of the CQ-$c\,c\,b\,b$ configuration, a set of centroids $C_i^\star \subset \mathbb{R}^c$ is learned independently for each channel group $i$ through the objective

$$C_i^\star = \underset{\substack{C \subset \mathbb{R}^c \\ |C|=2^b}}{\arg\min} \left\| A_{(ic-c+1):ic,\,:} - \text{cq}\left(A_{(ic-c+1):ic,\,:}\right) \right\|_F^2 \tag{5}$$

where $A_{(ic-c+1):ic,\,:}$ is the sub-matrix of $A$ containing all coupled channels from the $i$-th group, and cq quantizes each column vector to the nearest centroid in $C$ in terms of L2 distance. We use the k-means algorithm [26] with k-means++ initialization [1] to optimize the objective.

LLMs are more sensitive to the quantized precision of certain weights than others [20], hence centroids of CQ should be learned to bias towards preserving the precision of more important activations. To this end, we leverage an approximation to the Hessian to perform second-order-information-informed centroid learning. More concretely, we use the diagonals of the Fisher information matrix $\mathcal{F}$ to identify the more influential key/value activations and guide the centroid learning process. Using the diagonal Fisher information matrix for quantization was proposed in [22], and we extend it to the multi-channel case. For performing Fisher-guided centroid learning, we first save a key/value activation matrix $A$ and its gradient $g(A) = \frac{\partial}{\partial A}\mathcal{L}(A)$ on a calibration dataset, where $\mathcal{L}$ is the training loss function. We approximate the Hessian matrix using the diagonals of the Fisher information matrix, $\text{diag}(\mathcal{F}) = g(A) \odot g(A)$, which is the element-wise square of the gradient matrix. We use the sum of diagonal entries of the Fisher information matrix as a measure of importance for each group of activations, and obtain the centroid set $C_i^\star$ for the $i$-th channel group using the objective

$$C_i^\star = \underset{\substack{C \subset \mathbb{R}^c \\ |C|=2^b}}{\arg\min} \sum_j \underbrace{g\left(A_{(ic-c+1):ic,\,j}\right)^\top g\left(A_{(ic-c+1):ic,\,j}\right)}_{\text{partial sum of diag}(\mathcal{F})} \left\| A_{(ic-c+1):ic,\,j} - \text{cq}(A_{(ic-c+1):ic,\,j}) \right\|_2^2 \tag{6}$$

which we leverage weighted k-means to optimize. We discuss the overhead of centroid learning and centroid storage in Section E in the appendix.

### 3.3 Efficient Inference Through Kernel Fusion

We design fused GPU kernels to enable efficient inference of CQ. During inference, dequantizing couple-quantized KV cache requires many random accesses for lookups of multi-dimensional centroids. If the centroids reside in GPU global memory, these random accesses would greatly hinder the inference efficiency. We circumvent this issue by caching centroids in the shared memory, which has significantly lower latency and higher bandwidth than global memory. Due to limited size of the shared memory for each thread block, we assign the work of a single channel group to each thread block, which only requires loading a single group of centroids into a block of shared memory. We perform kernel fusion to merge dequantization of key cache, positional encoding and KQ multiplication, as well as to merge dequantization of value cache and its multiplication with attention scores. We validate the inference efficiency of CQ empirically in Section 4.4.

## 4 Experiments

In this section, we perform extensive experiments to validate the effectiveness of our proposed CQ approach for KV cache compression. We first introduce the experimental setups including hardware, software, datasets, metrics, and baselines used. Then, we present the detailed empirical results and provide discussions. Finally, we perform an ablation study to validate the effectiveness of each component of our proposal.

**Hardware and Software** Experiments are performed on a Linux server equipped with 4 NVIDIA A100 40GB GPUs. Our software implementation of CQ is based on PyTorch [29] and the Hugging-Face Transformers library [39].

**Evaluation Metrics and Datasets** We compare different KV cache quantization by evaluating the quality of 5 LLMs on various benchmarks. The 5 LLMs considered are 1. LLaMA-7b, 2. LLaMA-13b

Table 1: Perplexity of LLMs on WikiText-2 under different KV cache quantization methods at varying bit widths. The results of INT, NF, and KVQuant (except -1b and -1b+1% sparse) are from [15]. "NaN" means Not a Number, which is caused by quantization numerical instability. Our proposed method CQ outperforms baselines under the same bit width.

| | Bits Per Activation | LLaMA-7b | LLaMA-13b | LLaMA-2-7b | LLaMA-2-13b | Mistral-7b |
|---|---|---|---|---|---|---|
| FP16 | 16 | 5.68 | 5.09 | 5.12 | 4.57 | 4.76 |
| INT4 | 4.00 | 5.98 | 5.32 | 5.66 | 5.01 | 4.97 |
| INT4-g128 | 4.16 | 5.77 | 5.16 | 5.32 | 4.71 | 4.82 |
| NF4 | 4.00 | 5.87 | 5.23 | 5.47 | 4.90 | 4.91 |
| NF4-g128 | 4.25 | 5.77 | 5.17 | 5.30 | 4.71 | 4.83 |
| KVQuant-4b | 4.00 | 5.73 | 5.15 | 5.18 | 4.63 | 4.81 |
| KVQuant-4b+1% sparse | 4.32 | **5.70** | **5.11** | **5.14** | **4.59** | **4.78** |
| CQ-2c8b | 4.00 | **5.70** | **5.11** | **5.14** | **4.59** | 4.79 |
| INT2 | 2.00 | 11779 | 69965 | 4708 | 3942 | 573 |
| INT2-g128 | 2.14 | 37.37 | 41.77 | 117.88 | 93.09 | 51.96 |
| NF2 | 2.00 | 3210.5 | 5785.6 | 13601 | 4035.6 | 902.51 |
| NF2-g128 | 2.25 | 351.23 | 141.19 | 634.59 | 642.44 | 252.85 |
| KVQuant-2b | 2.00 | 8.17 | 7.29 | 9.75 | 29.25 | 7.33 |
| KVQuant-2b+1% sparse | 2.32 | 6.06 | 5.40 | 5.50 | 4.92 | 5.16 |
| CQ-4c8b | 2.00 | 5.97 | 5.32 | 5.42 | 4.81 | 5.11 |
| CQ-4c9b | 2.26 | **5.88** | **5.26** | **5.32** | **4.74** | **4.98** |
| KVQuant-1b | 1.00 | 321.58 | 1617.40 | NaN | 4709.83 | 203.73 |
| KVQuant-1b+1% sparse | 1.32 | 9.93 | 7.97 | 9.50 | 13.76 | 10.07 |
| CQ-8c8b | 1.00 | 8.09 | 7.02 | 7.75 | 6.55 | 7.25 |
| CQ-8c10b | 1.27 | **6.78** | **6.00** | **6.25** | **5.47** | **5.90** |

[36], 3. LLaMA-2-7b, 4. LLaMA-2-13b [37], 5. Mistral-7b [18]. We evaluate the quality of LLMs using the metric perplexity on 2 datasets: WikiText-2 [27] and C4 [31], and zero-shot accuracy on 3 benchmarks: WinoGrande [32], PIQA [3], and ARC Challenge (Arc-C) [5]. Furthermore, we evaluate on long-context benchmarks GSM8K [6] with chain-of-thought (CoT), and few-shot MMLU [13] with CoT. For perplexity and accuracy evaluations, the KV cache of all tokens in all layers is quantized, during both the prefill and decoding stages. The experimental details, including the procedures for perplexity and benchmark evaluations, are presented in Section A in the Appendix. More experimental results, including a comparison between CQ and KIVI [25] on LongBench [2], and results on more models and passkey retrieval [28], can be found in the Appendix.

**Baselines** We compare our proposed approach with uncompressed FP16 KV cache and competitive KV cache quantization methods, including 1. uniform integer (INT) quantization (without grouping and with a group size of 128), 2. NormalFloat (NF) quantization [8] (without grouping and with a group size of 128), 3. KVQuant [15] (dense-only and with 1% outliers stored in sparse format). KVQuant-b+1% sparse is a dense-and-sparse method that stores outlier activations in a sparse matrix and requires an additional sparse matrix multiplication during inference, which introduces extra computational overhead. For calibration, we use the same set of 16 sequences of WikiText-2, each with 2048 tokens, for KVQuant and CQ. Other methods do not require calibration. Calibration is performed only once and the learned centroids are used for all downstream evaluations. Calibration is done on the training set of WikiText-2, while perplexity and accuracy are evaluated on the test sets of different datasets and benchmarks. For 1-bit and 2-bit KVQuant, we employ Q-Norm to mitigate distribution shift, as recommended by [15]. We report Bits Per Activation (BPA) to measure the compression rate of each method, where each activation in the uncompressed KV cache is a 16-bit float. Detailed calculations of bits per activation for CQ are presented in Section F in the appendix.

## 4.1 Results

Table 1 presents the perplexity of LLMs on WikiText-2 under different KV quantization methods. CQ consistently outperforms baselines under the same quantization bit width. In low bit width regions of 1-bit and 2-bit quantization, dense-only quantization baselines quickly deteriorates in quality while CQ preserves quality well. We highlight that CQ-8c8b (1 bit per activation) outperforms KVQuant-2b (2 bits per activation) with only half the memory. CQ also compares favorably against

Table 2: Accuracy of LLMs on 3 benchmarks under different KV cache quantization methods at varying bit widths.

| | Bits Per Activation | Benchmark | LLaMA-7b | LLaMA-13b | LLaMA-2-7b | LLaMA-2-13b | Mistral-7b | Average |
|---|---|---|---|---|---|---|---|---|
| FP16 | 16 | WinoGrande | 69.93 | 72.69 | 68.90 | 71.98 | 73.88 | 65.55 |
| | | PIQA | 78.67 | 79.16 | 78.07 | 79.16 | 80.58 | |
| | | ARC-C | 41.72 | 46.42 | 43.43 | 48.29 | 50.34 | |
| KVQuant-4b | 4.00 | WinoGrande | 69.53 | 72.61 | 67.96 | 71.59 | **73.88** | 65.17 |
| | | PIQA | **78.62** | **79.22** | 77.86 | 78.94 | 80.58 | |
| | | ARC-C | **42.32** | 45.99 | 42.75 | 46.67 | 49.06 | |
| KVQuant-4b+1% sparse | 4.32 | WinoGrande | **70.72** | **73.40** | **68.67** | 72.30 | 73.72 | 65.57 |
| | | PIQA | 78.40 | 79.16 | **78.07** | **79.27** | **80.74** | |
| | | ARC-C | 41.38 | **46.76** | 43.17 | 47.87 | 49.91 | |
| CQ-2c8b | 4.00 | WinoGrande | 70.40 | 72.45 | 68.27 | **72.53** | 73.48 | 65.31 |
| | | PIQA | 78.61 | 79.11 | 77.91 | 78.62 | 80.52 | |
| | | ARC-C | 41.55 | 45.99 | **43.34** | 47.78 | 49.15 | |
| KVQuant-2b+1% sparse | 2.32 | WinoGrande | 68.03 | **71.43** | 67.64 | 70.17 | **70.80** | 63.79 |
| | | PIQA | **77.69** | 78.51 | 76.60 | **78.51** | **79.65** | |
| | | ARC-C | 38.74 | **45.14** | 41.47 | 44.97 | 47.53 | |
| CQ-4c9b | 2.26 | WinoGrande | **68.51** | 69.93 | 67.40 | **71.67** | 70.71 | 63.75 |
| | | PIQA | 76.82 | 78.51 | 77.09 | 77.31 | 79.48 | |
| | | ARC-C | **39.16** | 45.14 | 41.64 | 44.97 | **47.95** | |
| KVQuant-2b | 2.00 | WinoGrande | 53.59 | 59.35 | 51.70 | 51.30 | 63.46 | 52.20 |
| | | PIQA | 72.47 | 74.81 | 63.38 | 65.40 | 75.46 | |
| | | ARC-C | 32.00 | 34.47 | 22.44 | 24.66 | 38.57 | |
| CQ-4c8b | 2.00 | WinoGrande | **67.48** | **70.72** | **66.45** | **69.06** | **69.38** | 62.85 |
| | | PIQA | **76.11** | **78.29** | **76.12** | **77.42** | **79.49** | |
| | | ARC-C | **38.48** | **44.03** | **39.93** | **44.11** | **45.65** | |
| KVQuant-1b+1% sparse | 1.32 | WinoGrande | 56.67 | 61.01 | 57.77 | 57.30 | 58.17 | 54.31 |
| | | PIQA | 71.38 | 75.46 | 69.91 | 70.89 | 73.83 | |
| | | ARC-C | 29.69 | 35.32 | 31.48 | 32.59 | 33.19 | |
| CQ-8c10b | 1.27 | WinoGrande | **60.46** | **65.27** | **59.19** | **62.98** | **63.93** | 57.95 |
| | | PIQA | **73.45** | **75.90** | **73.07** | **74.37** | **77.31** | |
| | | ARC-C | **33.28** | **37.12** | **34.64** | **38.74** | **39.59** | |
| KVQuant-1b | 1.00 | WinoGrande | 50.51 | 48.46 | 50.91 | 49.41 | 49.80 | 41.35 |
| | | PIQA | 53.26 | 53.54 | 53.37 | 50.92 | 54.73 | |
| | | ARC-C | 21.76 | 21.33 | 20.65 | 21.67 | 19.88 | |
| CQ-8c8b | 1.00 | WinoGrande | **56.51** | **61.56** | **55.01** | **57.14** | **58.25** | 54.36 |
| | | PIQA | **71.16** | **73.99** | **71.22** | **73.01** | **75.24** | |
| | | ARC-C | **30.20** | **33.79** | **30.20** | **34.30** | **33.79** | |

Table 3: Accuracy of LLaMA-2-7b on 5 long-context benchmarks under different KV cache quantization methods at varying bit widths.

| | Bits Per Activation | GSM8K, CoT | MMLU, CoT Fewshot | | | |
|---|---|---|---|---|---|---|
| | | | STEM | Humanities | Social | Other |
| FP16 | 16 | 13.57 | 33.43 | 41.12 | 50.74 | 56.60 |
| KVQuant-4b+1% sparse | 4.32 | 14.33 | 31.04 | 41.12 | **48.37** | 55.43 |
| CQ-2c8b | 4.00 | **14.71** | **33.73** | **43.44** | 47.77 | **56.01** |
| KVQuant-2b+1% sparse | 2.32 | **10.31** | **28.06** | 35.64 | 42.43 | **46.39** |
| CQ-4c9b | 2.26 | **10.31** | 27.76 | **35.91** | **44.51** | 45.75 |
| KVQuant-2b | 2.00 | 2.27 | 9.85 | 12.55 | 20.18 | 19.94 |
| CQ-4c8b | 2.00 | **8.04** | **25.67** | **30.89** | **45.40** | **41.94** |
| KVQuant-1b+1% sparse | 1.32 | 2.27 | 10.75 | 14.09 | 20.77 | 19.94 |
| CQ-8c10b | 1.27 | **2.35** | **13.13** | **21.81** | **28.19** | **26.98** |
| KVQuant-1b | 1.00 | 0.68 | 0.00 | 0.00 | 0.00 | 0.00 |
| CQ-8c8b | 1.00 | **1.74** | **5.37** | **11.39** | **20.77** | **16.72** |

dense-and-sparse baselines by outperforming KVQuant+1% sparse in most cases despite using less bits. We present the perplexity results on C4 in Table 7 in the appendix. Table 2 presents the accuracy results of KVQuant and CQ on different benchmarks. CQ consistently outperforms dense-only KVQuant at 1-bit and 2-bit, and performs better or on par with the dense-and-sparse KVQuant under the same bit width. Table 3 presents the accuracy comparison of KVQuant and CQ on long-context benchmarks, with CoT or few-shot CoT. CQ mostly outperforms KVQuant under similar bit widths.

Table 4: Accuracy of LLaMA-2 models with couple-quantized KV cache and a sliding window of 32 recent tokens cached in FP16. CQ achieves minimal accuracy degradation compared to the FP16 baseline.

|  | Quant. | BPA | WinoGrande | PIQA | Arc-C | Arc-E | Hellaswag | Average |
|---|---|---|---|---|---|---|---|---|
| LLaMA-2-7b | FP16 | 16 | 68.90 | 78.07 | 43.43 | 76.30 | 57.14 | 64.768 |
|  | CQ-2c8b | 4.00 | 69.14 | 78.18 | 43.34 | 76.52 | 57.12 | 64.860 (+0.092) |
|  | CQ-4c8b | 2.00 | 69.06 | 77.86 | 42.83 | 76.01 | 56.79 | 64.510 (-0.258) |
|  | CQ-8c8b | 1.00 | 69.14 | 77.91 | 42.92 | 75.67 | 55.11 | 64.150 (-0.618) |
| LLaMA-2-13b | FP16 | 16 | 71.98 | 79.16 | 48.29 | 79.42 | 60.04 | 67.778 |
|  | CQ-2c8b | 4.00 | 72.30 | 78.94 | 47.95 | 79.55 | 60.18 | 67.784 (+0.006) |
|  | CQ-4c8b | 2.00 | 72.30 | 78.89 | 47.61 | 79.21 | 59.84 | 67.570 (-0.208) |
|  | CQ-8c8b | 1.00 | 72.22 | 78.84 | 47.78 | 79.12 | 58.55 | 67.302 (-0.476) |

## 4.2 Near-native Performance with Sliding Window Full-precision Cache

CQ mostly outperforms competitive baselines under the same bit width for fully quantized KV cache, and we further investigate the performance of CQ when combined with a sliding window of recent tokens cached in FP16 precision. This sliding window of full-precision cache only introduces a small constant memory overhead for each sequence. In Table 4, we present the accuracy of LLaMA-2 models on 5 benchmarks (WinoGrande, PIQA, Arc-C, Arc Easy (Arc-E) [5], and Hellaswag [42]) with a sliding window of 32 recent tokens in FP16 and the rest of the tokens coupled-quantized. CQ preserves the accuracy well, achieving a mere 0.476–0.618% loss in average accuracy over 5 tasks with 1-bit quantization. CQ also preserves perplexity well at 1-bit with a sliding window of 128 tokens in FP16, as shown in Figure 1.

Table 5: Ablative study on the effects of channel coupling for quantizing keys only, values only, and both keys and values, using 1-bit CQ. Perplexity of LLaMA-7b on WikiText-2 is reported.

|  | BPA | Keys Only | Values Only | Keys & Values |
|---|---|---|---|---|
|  |  | | Perplexity ↓ | |
| CQ-1c1b | 1.00 | 17.17 | 177.13 | 620.08 |
| CQ-2c2b | 1.00 | 8.29 | 9.49 | 28.39 |
| CQ-4c4b | 1.00 | 7.10 | 7.11 | 10.47 |
| CQ-8c8b | 1.00 | 6.53 | 6.54 | 8.09 |
| FP16 | 16 | | 5.68 | |

Table 6: Ablative study on the effects of Fisher-guided centroid learning for CQ. Perplexity of LLaMA-7b on WikiText-2 is reported.

|  | Centroids | Perplexity ↓ |
|---|---|---|
| CQ-2c8b | Uniform | 5.77 |
|  | Fisher-guided | 5.70 |
| CQ-4c8b | Uniform | 6.86 |
|  | Fisher-guided | 5.97 |
| CQ-8c8b | Uniform | 32.12 |
|  | Fisher-guided | 8.09 |

## 4.3 Ablation Study

We perform a set of ablation experiments to answer the following questions. The experimental results are presented in Tables 5 & 6.

**Q1: Under the same quantization bit width, does coupling more channels lead to better model quality?** Yes, the model performance approaches the FP16 baseline level as the number of coupled channels increases, as shown in Table 5. This also holds true for different models and bit widths, as shown in Figure 1 and Table 8 in the appendix.

**Q2: Is channel coupling effective for quantizing both keys and values?** Yes, channel coupling is highly effective for both keys and values. As shown in Table 5, perplexity improves significantly under the same bit width as the number of coupled channels increases, and holds true for both key-only and value-only quantization.

**Q3: Does Fisher-guided learning of centroids produce better model quality than uniform clustering?** Yes, Fisher-guided centroid learning improves the model perplexity over uniform clustering, as shown in Table 6. Table 8 in the appendix further shows it is effective for different models.

## 4.4 Efficiency of CQ

We study the efficiency of CQ by comparing its decoding throughput with uncompressed FP16 KV cache using the HuggingFace Transformers implementation [39]. We measure the decoding throughput by running LLaMA-2-7b on an A100-40GB GPU with CQ and FP16 KV cache to process a 100-token prompt and generate 1000 tokens. We increase the inference batch size until the GPU runs out of memory. Results are presented in Figure 4. CQ achieves 3.75×, 7.5×, and 15× larger batch size than the FP16 baseline at 4-bit, 2-bit, 1-bit quantization, respectively. Moreover, CQ improves the decoding throughput by 1.4–3.5× relative to the FP16 baseline. Additional latency measurements for CQ are presented in Section K in the Appendix.

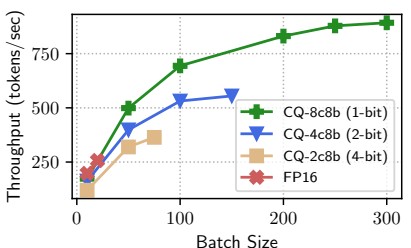

Figure 4: Inference throughput of CQ versus FP16 KV cache for LLaMA-2-7b.

# 5 Related Works

Existing works mitigate the high memory overhead of KV cache through token eviction, quantization, or tensor offloading. Scissorhands [24] and H2O [47] achieve KV cache compression while preserving model quality by storing only the pivotal tokens and evicting the unimportant ones. KIVI [25] proposes to quantize keys per-channel and values per-token using group-wise integer quantization. KVQuant [15] proposes sensitivity-based and dense-and-sparse quantization for KV cache to reduce quantization errors. Flexgen [34] proposes to offload KV cache instead of weights to enable high-throughput inference on a single GPU. Weight quantization [11, 23, 44] is an orthogonal line of work that reduce GPU memory requirements and improve effciency of LLM inference and fine-tuning [45]. FlashAttention [7] and NoMAD-Attention [46] are system optimizations for speeding up LLM inference on GPUs and CPUs, respectively. Product quantization [17] is a method for nearest neighbor search that compresses vectors by decomposing the vector space into a Cartesian product of low-dimensional subspaces, and quantizing each subspace independently.

# 6 Conclusion

We propose Coupled Quantization (CQ) for mitigating the latency and throughput bottleneck of LLM inference by quantizing KV cache. We discover that channels of KV cache are highly interdependent, which implies the existing approach of per-channel quantization approach is sub-optimal. We propose channel coupling to exploit the interdependency across channels to achieve more information-efficient encoding of key/value activations. Extensive experiments demonstrate that our method outperforms competitive baselines in model quality in most cases, and can reasonably preserve model quality with KV cache quantized down to 1 bit.

## Acknowledgements

This work was supported by National Science Foundation SHF-2211815, Ken Kennedy Institute, and grants from Adobe and VMware.

## Limitations & Broader Impacts

KV cache quantization is a form of lossy compression which inevitably affects model quality. Although we study its effects on perplexity and accuracy, it remains unclear how it affects other aspects of the model such as hallucination and adversarial robustness. Making LLM inference more efficient contributes to the democratization of artificial intelligence and the reduction of carbon footprints. We expect no additional negative societal impacts other than the ones already posed by LLMs.

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

# Appendix / Supplemental Material

## A    Experimental Details

**Perplexity Evaluations**    Perplexity is evaluated on the test set of the datasets, WikiText-2 and C4, at the maximum context length of the LLMs (2048 for LLaMA, 4096 for LLaMA-2, and 8192 for Mistral), following the setup in [15].

**Benchmark Evaluations**    We use `lm-evaluation-harness` [12] (package version 0.4.2) for performing benchmark evaluations.  We use the following task names: `winogrande`, `piqa`, `arc_challenge`, `gsm8k_cot`, `mmlu_flan_cot_fewshot_humanities`, `mmlu_flan_cot_fewshot_stem`, `mmlu_flan_cot_fewshot_social_sciences`, `mmlu_flan_cot_fewshot_other`.

## B    Perplexity Results on C4

Table 7 presents the perplexity results of different quantization algorithms on the test set of C4 dataset. Our proposed method CQ performs better than or on par with baselines under the same bit width.

Table 7: Perplexity of LLMs on C4 under different KV cache quantization methods at varying bit widths. The results of INT, NF, and KVQuant (except -1b and -1b+1% sparse) are from [15].

| | Bits Per Activation | LLaMA-7b | LLaMA-13b | LLama-2-7b | LLaMA-2-13b | Mistral-7b |
|---|---|---|---|---|---|---|
| FP16 | 16 | 7.08 | 6.61 | 6.63 | 6.05 | 5.71 |
| INT4 | 4.00 | 7.40 | 6.82 | 7.31 | 6.59 | 5.91 |
| INT4-g128 | 4.16 | 7.16 | 6.67 | 6.87 | 6.20 | 5.76 |
| NF4 | 4.00 | 7.27 | 6.74 | 7.09 | 6.45 | 5.85 |
| NF4-g128 | 4.25 | 7.16 | 6.66 | 6.86 | 6.20 | 5.77 |
| KVQuant-4b | 4.00 | 7.13 | 6.65 | 6.70 | 6.11 | 5.75 |
| KVQuant-4b+1% sparse | 4.32 | **7.09** | **6.62** | **6.65** | **6.06** | **5.72** |
| CQ-2c8b | 4.00 | 7.11 | 6.64 | 6.67 | 6.09 | 5.74 |
| INT2 | 2.00 | 10892 | 100870 | 4708 | 4220 | 477 |
| INT2-g128 | 2.14 | 43.49 | 56.25 | 113.49 | 97.04 | 50.73 |
| NF2 | 2.00 | 2850.1 | 4680.3 | 13081.2 | 4175.6 | 1102.3 |
| NF2-g128 | 2.25 | 248.32 | 118.18 | 420.05 | 499.82 | 191.73 |
| KVQuant-2b | 2.00 | 10.28 | 9.05 | 15.16 | 43.77 | 8.40 |
| KVQuant-2b+1% sparse | 2.32 | 7.38 | **6.83** | 7.06 | 6.38 | 6.08 |
| CQ-4c8b | 2.00 | 7.52 | 6.96 | 7.23 | 6.52 | 6.17 |
| CQ-4c9b | 2.26 | **7.37** | 6.84 | **7.02** | **6.36** | **5.99** |
| KVQuant-1b | 1.00 | 168.90 | 1316.41 | 362.94 | 4223.37 | 127.07 |
| KVQuant-1b+1% sparse | 1.32 | 11.18 | 9.56 | 16.04 | 22.87 | 10.53 |
| CQ-8c8b | 1.00 | 12.13 | 10.53 | 12.49 | 10.53 | 9.89 |
| CQ-8c10b | 1.27 | **9.12** | **8.23** | **9.03** | **8.01** | **7.46** |

## C    Effects of Channel-coupling and Fisher-guided Centroid Learning

We validate the effectiveness of our proposed channel-coupling and Fisher-guided centroid learning by compressing LLaMA-7b KV cache to 1-bit and 2-bit, and present the perplexity results and quantization errors ($\|A - \mathrm{cq}(A)\|_F^2$ averaged over layers) on WikiText-2 under different CQ configurations in Figure 5. As the number of coupled channels increases, perplexity improves significantly and approaches the FP16 performance. The quantization errors of keys and values also decrease as the number of coupled channels increase. Although Fisher-guided centroid learning increases the quantization error, it better preserves the salient activations and achieves lower perplexity.

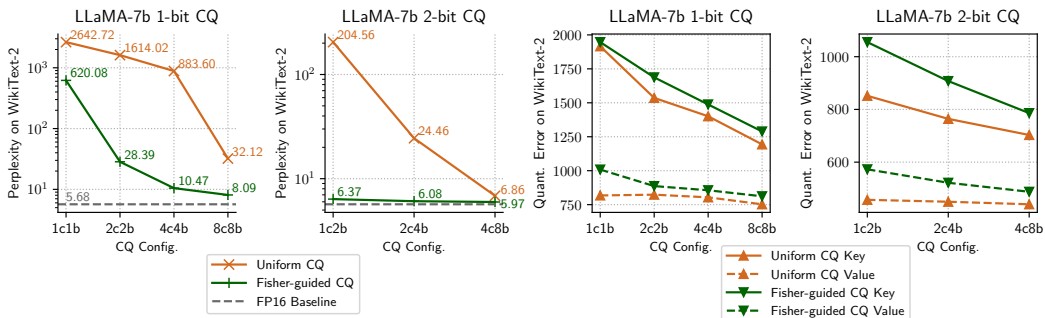

Figure 5: Perplexity and key/value quantization errors (averaged over all layers) of LLaMA-7b on WikiText-2. Channels coupling and Fisher-guided centroid learning are effective for improving perplexity.

## D Ablation Study on More Models

We study the effectiveness of each component of our proposed approach in Table 8. We evaluate the perplexity of 2 models Mistral-7b and LLaMA-2-13b on WikiText-2 using CQ at 2 bits per activation, with varying number of channels coupled and comparing uniform centroid learning and Fisher-guided centroid learning. Fisher-guided centroids significantly improve model quality as demonstrated by lower perplexity. With either uniform centroids or Fisher-guided centroids, perplexity improves as the number of coupled channels increases. Hence, our proposed techniques of channel coupling and Fisher-guided centroid learning are both effective for maintaining model quality.

Table 8: Perplexity of different models on WikiText-2 using CQ with varying number of coupled channels and fisher-guided centroids. Perplexity consistently improves as the number of coupled channels increases.

|  | Mistral-7b | | | | | | LLaMA-2-13b | | | | | |
|---|---|---|---|---|---|---|---|---|---|---|---|---|
| Bits Per Activation | 2 | 2 | 2 | 2 | 2 | 2 | 2 | 2 | 2 | 2 | 2 | 2 |
| Num. of Channels Coupled | 1 | 2 | 4 | 1 | 2 | 4 | 1 | 2 | 4 | 1 | 2 | 4 |
| Fisher-guided Centroids | ✗ | ✗ | ✗ | ✓ | ✓ | ✓ | ✗ | ✗ | ✗ | ✓ | ✓ | ✓ |
| Perplexity ↓ | 97.76 | 16.29 | 5.42 | 5.34 | 5.20 | 5.11 | 890.42 | 171.96 | 6.62 | 6.06 | 4.91 | 4.81 |

## E Overhead of Centroid Learning and Storage

We present the computational overhead of centroid learning and the memory overhead of centroid storage for CQ in Table 9. The centroid learning process of CQ consists of many independent k-means runs, which can be time-consuming on CPUs. Hence, we leverage a GPU implementation to accelerate the learning process. In all our experiments, we use k-means++ initialization and run 100 iterations of k-means on a single GPU to obtain the centroids. The memory overhead of storing the centroids can be calculated as $l \times 2 \times h \times d \times 2^b$ FP16 numbers, where $l$ is the number of layers, 2 is for keys and values, $h$ is the number of key/value attention heads, $d$ is the number of channels in a single-head key/value activation embedding, and $b$ is the bit width of quantized codes. As shown in Table 9, CQ can easily scale to large model sizes with low learning and memory overheads.

## F Bits Per Activation Calculations for CQ

To calculate bits per activation, we assume the batch size is 16 and the sequence length is 65,536. For CQ-$c$c$b$b, the storage of quantized codes requires the following number of bits

$$16 \times 65536 \times l \times 2 \times h \times d \times b/c$$

where $l$ is the number of layers, 2 is for key and value, $h$ is the number of key/value attention heads, $d$ is the number of channels for a single head of attention, and $b/c$ is the average number of bits per token per channel.

Table 9: Learning and memory overhead of different CQ configurations and models. The number of centroid parameters are shown in millions, and the percentage to the model weights is shown in brackets.

| CQ Config. | Centroid Learning Time | | | | | Parameter Count in Centroids | | | | |
| | 2c8b | 4c8b | 4c9b | 8c8b | 8c10b | 2c8b | 4c8b | 4c9b | 8c8b | 8c10b |
| --- | --- | --- | --- | --- | --- | --- | --- | --- | --- | --- |
| LLaMA-7b | 53 mins | 28 mins | 62 mins | 14 mins | 104 mins | 67.11M (0.996%) | 67.11M (0.996%) | 134.22M (1.992%) | 67.11M (0.996%) | 268.44M (3.984%) |
| LLaMA-13b | 94 mins | 44 mins | 96 mins | 22 mins | 160 mins | 104.86M (0.806%) | 104.86M (0.806%) | 209.72M (1.612%) | 104.86M (0.806%) | 419.43M (3.224%) |
| LLaMA-2-7b | 54 mins | 28 mins | 62 mins | 14 mins | 104 mins | 67.11M (0.996%) | 67.11M (0.996%) | 134.22M (1.992%) | 67.11M (0.996%) | 268.44M (3.984%) |
| LLaMA-2-13b | 83 mins | 44 mins | 96 mins | 23 mins | 162 mins | 104.86M (0.806%) | 104.86M (0.806%) | 209.72M (1.612%) | 104.86M (0.806%) | 419.43M (3.224%) |
| Mistral-7b | 13 mins | 7 mins | 15 mins | 4 mins | 27 mins | 16.78M (0.232%) | 16.78M (0.232%) | 33.56M (0.464%) | 16.78M (0.232%) | 67.11M (0.928%) |

The learned centroids of CQ are stored in FP16 format. Hence the storage of CQ centroids requires the following number of bits

$$l \times 2 \times h \times d \times 2^b \times 16$$

CQ has no other storage overhead besides quantized codes and centroids. Therefore, the average bits per activation for CQ-$c$c$b$b is calculated as

$$\frac{16 \times 65536 \times l \times 2 \times h \times d \times b/c + l \times 2 \times h \times d \times 2^b \times 16}{16 \times 65536 \times l \times 2 \times h \times d \times 16} \times 16$$

$$= \frac{65536 \times b/c + 2^b}{65536}$$

# G   Generalizability of Learned Centroids

Centroids for CQ are learned once on a calibration dataset and can be used for different downstream tasks. We perform an ablation study to examine the effects of calibration data on the quality of learned CQ centroids. We use 16 sequences of 2048 tokens from different datasets as the calibration set for CQ and evaluate on 4 downstream tasks, and the results are presented in Table 10. Despite using different calibration datasets, CQ performs similarly in various downstream tasks. The results suggest that calibration on language modeling datasets, such as WikiText-2 and C4, provides transferable performance on downstream tasks.

Table 10: Zero-shot accuracy of LLaMA-2-7b with couple-quantized KV cache, calibrated using two different datasets WikiText-2 and C4. CQ displays similar accuracy in various downstream tasks, despite using different calibration datasets.

| | Calibration Dataset | Downstream Task | | | |
| | | WinoGrande | PIQA | Arc-C | GSM8K CoT |
| --- | --- | --- | --- | --- | --- |
| CQ-2c8b | WikiText-2 | 68.27 | 77.91 | 43.34 | 14.71 |
| | C4 | 68.35 | 77.86 | 43.16 | 14.71 |
| CQ-4c8b | WikiText-2 | 66.45 | 76.12 | 39.93 | 8.04 |
| | C4 | 66.22 | 76.61 | 39.93 | 8.34 |
| CQ-8c8b | WikiText-2 | 55.01 | 71.22 | 30.20 | 1.74 |
| | C4 | 56.27 | 71.55 | 30.52 | 1.90 |

Table 11: Accuracy comparison of CQ and KIVI on LongBench for LLaMA-2-7b. The results of KIVI are from [25].

| | Sliding Window Size | Qasper | QMSum | MultiNews | TREC | TriviaQA | SAMSum | LCC | RepoBench-P |
| --- | --- | --- | --- | --- | --- | --- | --- | --- | --- |
| FP16 | - | 9.52 | 21.28 | 3.51 | 66.00 | 87.72 | 41.69 | 66.66 | 59.82 |
| KIVI-2b | 32 | 9.26 | 20.53 | 0.97 | **66.00** | 87.42 | **42.61** | 66.22 | 59.67 |
| CQ-4c8b | 32 | **9.58** | **20.87** | **1.93** | **66.00** | **87.72** | 41.13 | **66.57** | **59.75** |

## H   Comparison with KIVI

We perform an empirical comparison of CQ with KIVI [25] using LLaMA-2-7b on LongBench [2]. For both methods, we use 2-bit quantization and a sliding-window full-precision cache of 32 tokens. We compare CQ-4c8b against KIVI with a group size of 32. The accuracy results are presented in Table 11. CQ mostly outperforms KIVI in preserving model accuracy.

## I   Passkey Retrieval

We compare the quantization quality of CQ and KVQuant by evaluating them on the passkey retrieval task [28]. We follow the setup in [15], and measure the success rate of passkey retrieval for LLaMA-2-7b at its maximum context length of 4096. Table 12 presents the passkey retrieval results, where CQ outperforms or performs the same as KVQuant at various bit widths.

Table 12: The passkey retrieval success rate of CQ and KVQuant at different quantization bit widths, for LLaMA-2-7b at its maximum context length of 4096.

|  | Bit Width | Success Rate |
|---|---|---|
| KVQuant-4b+1% sparse | 4.32 | **100%** |
| KVQuant-4b | 4.00 | **100%** |
| CQ-2c8b | 4.00 | **100%** |
| KVQuant-2b+1% sparse | 2.32 | 94% |
| CQ-4c9b | 2.26 | **98%** |
| KVQuant-2b | 2.00 | 0% |
| CQ-4c8b | 2.00 | **96%** |
| KVQuant-1b+1% sparse | 1.32 | 2% |
| CQ-8c10b | 1.27 | **78%** |
| KVQuant-1b | 1.00 | 0% |
| CQ-8c8b | 1.00 | **12%** |

## J   Results on LLaMA 3

We evaluate the effectiveness of CQ and KVQuant for quantizing the KV cache of the LLaMA 3 model [10]. For both methods, we quantize the KV cache of all tokens of the LLaMA-3-8b model, and perform the calibration on 16 sequences from the training set of WikiText-2 and evaluate perplexity on the test set, and benchmark on 3 downstream tasks: WinoGrande, PIQA and Arc Challenge. The results are presented in Table [10]. CQ mostly outperforms KVQuant, especially in lower bit widths.

Table 13: The effectiveness of CQ and KVQuant on LLaMA-3-8b.

|  | Bits Per Activation | WikiText-2 PPL ↓ | WinoGrande ↑ | PIQA ↑ | Arc-C ↑ |
|---|---|---|---|---|---|
| FP16 | 16 | 5.54 | 72.69 | 79.71 | 50.51 |
| KVQuant-4b | 4.00 | 5.66 | 72.77 | **79.98** | 47.44 |
| CQ-2c8b | 4.00 | **5.58** | **73.16** | 78.84 | **49.83** |
| KVQuant-2b | 2.00 | 18.96 | 56.27 | 63.49 | 24.40 |
| CQ-4c8b | 2.00 | **6.09** | **69.22** | **78.62** | **44.03** |
| KVQuant-1b | 1.00 | 22238.91 | 50.04 | 53.05 | 22.35 |
| CQ-8c8b | 1.00 | **9.56** | **56.04** | **72.58** | **32.51** |

## K   Latency Measurements

We measure the prefill and decoding latency of CQ, with and without a sliding window of full-precision cache, and compare against KIVI [25] and full-precision FP16 cache. The latency mea-

surements are conducted on a single A100-40GB GPU for LLaMA-2-7b with a batch size of 1 and a prompt length of 2000 tokens The results are presented in Table 14. CQ achieves prefill and decoding latency on par with KIVI.

Table 14: Latency measurements of decoding LLaMA-2-7b with a batch size of 1 and a prompt of 2000 tokens, averaged over 100 tokens, using FP16 KV cache, and CQ and KIVI quantized KV cache.

| Full-precision Sliding-window Length | Prefill Time (s) | Decoding Time (s) |
|---|---|---|
| FP16 | - | 0.853 | 0.0559 ± 0.0044 |
| KIVI-4b | 32 | 1.483 | 0.0693 ± 0.0212 |
| KIVI-2b | 32 | 1.291 | 0.0684 ± 0.0213 |
| CQ-2c8b | 0 | 1.820 | 0.0695 ± 0.0056 |
| CQ-2c8b | 32 | 1.926 | 0.0701 ± 0.0057 |
| CQ-4c8b | 0 | 1.684 | 0.0704 ± 0.0056 |
| CQ-4c8b | 32 | 1.790 | 0.0706 ± 0.0058 |
| CQ-8c8b | 0 | 1.726 | 0.0670 ± 0.0070 |
| CQ-8c8b | 32 | 1.857 | 0.0799 ± 0.0066 |

## L    Extreme Sub-1-bit Quantization

We demonstrate the effectiveness of CQ by performing extreme sub-1-bit quantization with LLaMA-7b. We evaluate the perplexity of CQ-16c12b, which use a 12-bit code for every group of 16 coupled channels, averaging 0.81 bits per activation. Table 15 presents the results with comparison to FP16 cache and KVQuant-1b. CQ preserves model quality reasonably under extreme KV cache compression.

Table 15: Perplexity of LLaMA-7b under extreme KV cache compression with CQ-16c12b (12 bits per 16 coupled channels, averaging 0.81 bits per activation).

| | BPA | WikiText-2 ↓ | C4 ↓ |
|---|---|---|---|
| FP16 | 16 | 5.68 | 7.08 |
| KVQuant-1b | 1.00 | 321.58 | 168.90 |
| CQ-16c12b | 0.81 | 8.71 | 14.40 |

## M    Correlation Matrices and Scatter Plots

Figure 6 and 7 show the correlation matrices for key and value channels of each layer of the LLaMA-7b model on 262k tokens of WikiText-2. Figure 8 and 9 present the 2D scatter plots of key and value channel pairs of 4 layers of the LLaMA-7b model on 262k tokens of WikiText-2.

## N    Variations in Channel Correlation

As shown in Figure 6 and 7, the amount of correlation between channels varies depending on the layer. In Table 16, we study this variation by presenting the mean absolute correlation (MAC), excluding the diagonals, for different layers of LLaMA-7b on 262K tokens of WikiText-2. Although the key correlation of the first few layers are significantly higher than the later layers, the key/value correlation of any layer is never very close to zero, meaning channel coupling can be effective for any layer. Furthermore, the ablation study presented in Table 5 also demonstrates the effectivenss of channel coupling for capturing the channel correlation.

## O    Limitations

Low-bit quantization is a type of lossy model compression, which affects the model quality including metrics such as perplexity and accuracy, and also impacts model safety including toxicity and bias [41]. This work does not study the effects of extreme low-bit quantization of KV cache on the safety aspects of the model.

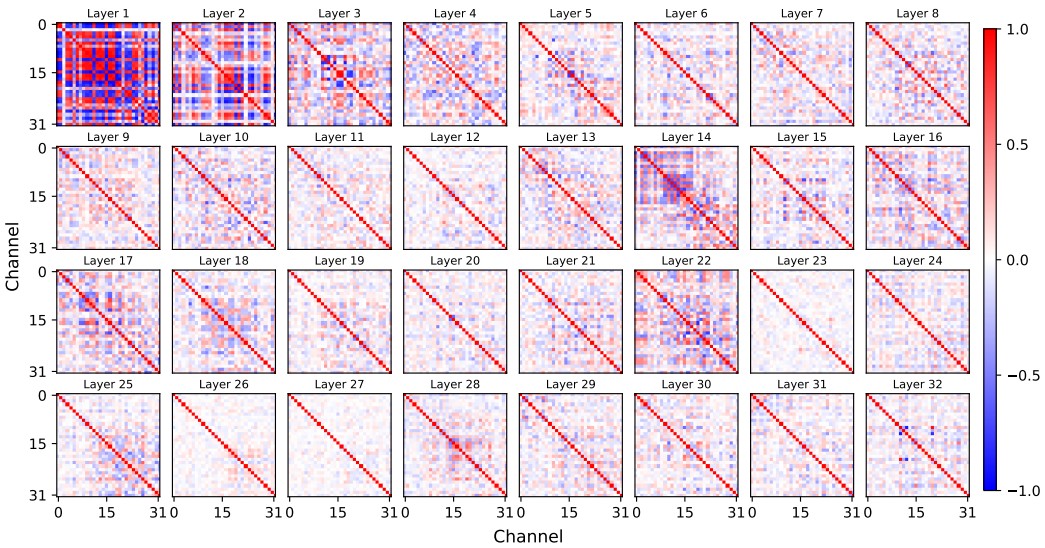

Figure 6: Correlation matrix for the first 32 channels of pre-RoPE **key** activation embeddings of all LLaMA-7b layers on WikiText-2.

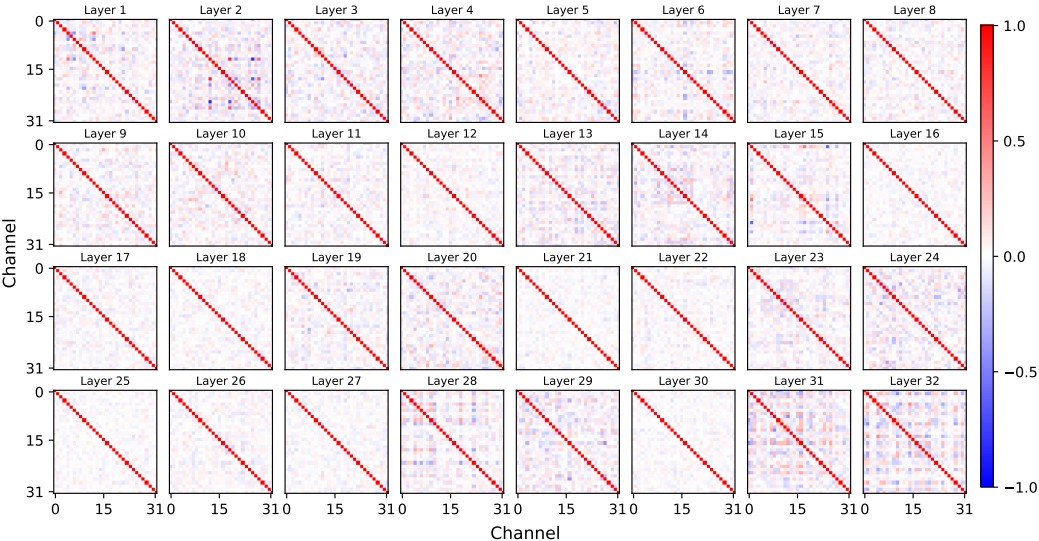

Figure 7: Correlation matrix for the first 32 channels of **value** activation embeddings of all LLaMA-7b layers on WikiText-2.

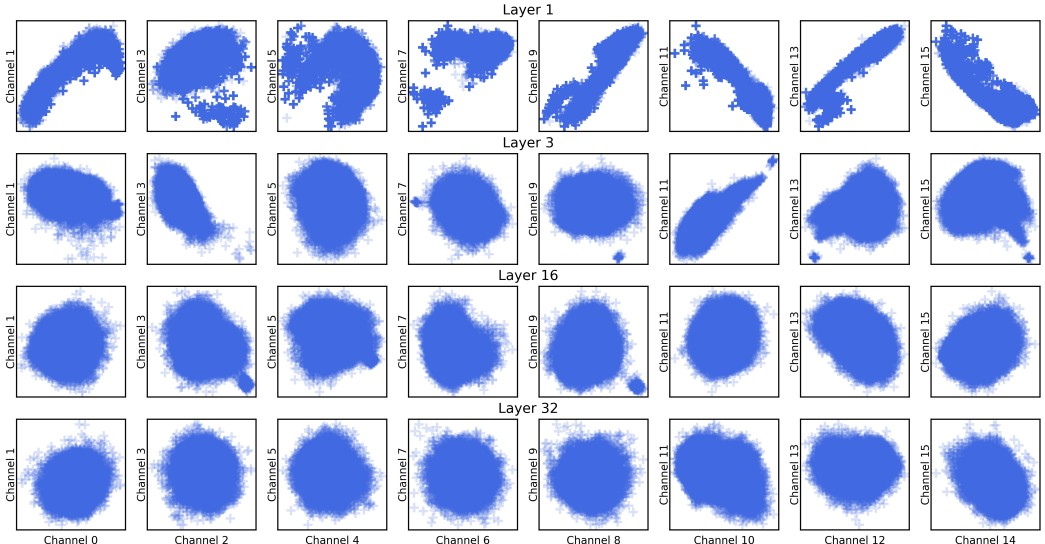

Figure 8: Scatter plots of pairs of channels in pre-RoPE **key** activation embeddings of 4 LLaMA-7b layers on WikiText-2.

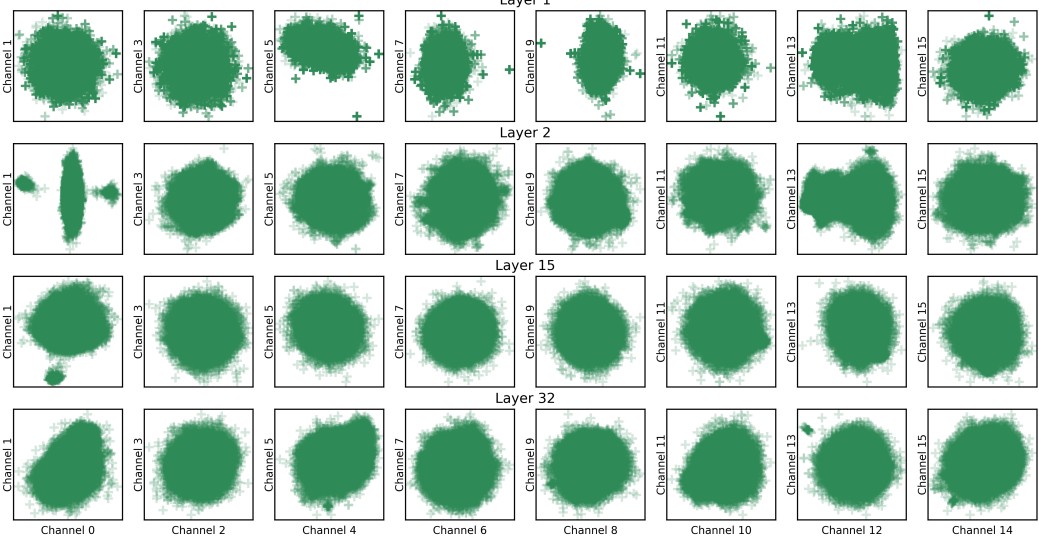

Figure 9: Scatter plots of pairs of channels in **value** activation embeddings of 4 LLaMA-7b layers on WikiText-2.

Table 16: The mean absolute correlation (MAC), excluding the diagonals, of key/value channels in different layers of LLaMA-7b, computed on 262K tokens of WikiText-2.

| Layer | 1 | 2 | 3 | 4 | 5 | 6 | 7 | 8 |
|---|---|---|---|---|---|---|---|---|
| Key | 0.407 | 0.212 | 0.193 | 0.178 | 0.114 | 0.113 | 0.115 | 0.122 |
| Value | 0.071 | 0.084 | 0.061 | 0.073 | 0.055 | 0.056 | 0.056 | 0.057 |

| Layer | 9 | 10 | 11 | 12 | 13 | 14 | 15 | 16 |
|---|---|---|---|---|---|---|---|---|
| Key | 0.131 | 0.138 | 0.090 | 0.098 | 0.094 | 0.141 | 0.109 | 0.114 |
| Value | 0.061 | 0.067 | 0.047 | 0.042 | 0.065 | 0.065 | 0.062 | 0.035 |

| Layer | 17 | 18 | 19 | 20 | 21 | 22 | 23 | 24 |
|---|---|---|---|---|---|---|---|---|
| Key | 0.136 | 0.111 | 0.091 | 0.101 | 0.102 | 0.156 | 0.071 | 0.094 |
| Value | 0.039 | 0.038 | 0.052 | 0.070 | 0.031 | 0.038 | 0.061 | 0.074 |

| Layer | 25 | 26 | 27 | 28 | 29 | 30 | 31 | 32 |
|---|---|---|---|---|---|---|---|---|
| Key | 0.107 | 0.069 | 0.057 | 0.103 | 0.095 | 0.097 | 0.100 | 0.090 |
| Value | 0.027 | 0.044 | 0.038 | 0.072 | 0.070 | 0.035 | 0.105 | 0.090 |

