# OpenReview forum: "KV Cache is 1 Bit Per Channel: Efficient Large Language Model Inference with Coupled Quantization"
_NeurIPS.cc/2024/Conference — NeurIPS 2024 poster_

### Official Review · Reviewer_rmbd · 2024-07-11

**Soundness:** 3
**Presentation:** 3
**Contribution:** 3
**Rating:** 6
**Confidence:** 5

**Summary:**

This paper presents a novel approach to KV Cache compression in Large Language Models (LLMs) called Coupled Quantization (CQ). The authors analyze the correlation between different channels in the KV Cache from an information entropy perspective, revealing significant interdependencies. Leveraging this insight, they propose a multi-channel joint non-uniform quantization method that achieves superior compression performance compared to previous per-channel quantization approaches. Experimental results demonstrate that by combining CQ with a sliding window approach, the KV Cache can potentially be compressed to 1 bit while maintaining model quality.

**Strengths:**

1. Thorough Analysis and Clear Motivation: The paper provides a detailed and intuitive explanation of the inter-channel correlation using information entropy, solidly justifying the design of Coupled Quantization. The motivation and effectiveness of the proposed method are well-articulated and supported by comprehensive experimental results.
﻿2. Good Performance at Low Bit-widths: CQ demonstratesadvantages over previous per-channel quantization methods without introducing additional overhead.
﻿3. Practical Efficiency Gains: The authors demonstrate substantial improvements in inference throughput and batch size compared to the FP16 baseline, highlighting the practical benefits of their approach.

**Weaknesses:**

1. Limited Evaluation Tasks: The main experiments focus on datasets (WinoGrande, PIQA, and ARC-C) that primarily consist of short-context QA or multiple-choice questions. Testing KV Cache quantization on these tasks may not fully demonstrate its benefits, as KV Cache compression is most relevant in long-text scenarios.
﻿2. Incomplete Explanation of Channel Correlation: While the visualizations in Figure 2 and the appendix show some layers have significant channel correlation, this is mainly evident in the first few layers. The paper doesn't fully explain the existence and variability of channel correlation across different layers, which may raise questions about the method's general applicability.

**Questions:**

1. Quantization Setting: Is the KV Cache quantized during the prefill stage, or only during the decoding stage (after full-precision prefill computation), similar to the KIVI approach?
2. Channel Grouping Strategy: The paper chooses to group adjacent continuous channel groups together. Would coupling non-adjacent channels with high mutual information potentially bring more benefits?
3. CUDA Kernel Implementation Questions: The compute pattern described for kernel fusion maybe potentially inefficient. Given the relatively small size of centroids for each channel group (e.g., 256 bytes for 4-bit quantization), is shared memory size truly a limiting factor? Could the efficiency be improved by adjusting the thread block assignments and reduction strategies for operations like $QK^T$ and $SV$?

**Limitations:**

Based on my review of the paper, I believe the authors have partially addressed limitations and potential negative societal impacts.

---

> ### Author Rebuttal · Authors · 2024-08-07
>
> We sincerely thank the reviewer for the insightful review and invaluable feedback. We address the reviewer's concerns as follows.
>
> **[W1] Limited Evaluation Tasks: KV Cache compression is most relevant in long-text scenarios.**
> - We appreciate the reviewer's suggestion to test KV cache quantization in long-text scenarios. We present additional experiments in long-text settings below. Specifically, we test CQ and KVQuant using Llama-2-7b on GSM8K with chain-of-thought (CoT) and MMLU with CoT Fewshot (gsm8k_cot, mmlu_flan_cot_fewshot_humanities, mmlu_flan_cot_fewshot_stem, mmlu_flan_cot_fewshot_social_sciences, mmlu_flan_cot_fewshot_other from lm-evaluation-harness).
> - In long-text settings, CQ mostly outperforms or performs similarly to KVQuant under the same bit width.
>
> |  | BPA | GSM8K CoT | MMLU (STEM) CoT Fewshot | MMLU (Humanities) CoT Fewshot | MMLU (Social Sciences) CoT Fewshot | MMLU (Other) CoT Fewshot |
> |---|---|---|---|---|---|---|
> | KVQuant-4b+1% sparse | 4.32 | 14.33 | 31.04 | 41.12 | **48.37** | 55.43 |
> | CQ-2c8b | 4.00 | **14.71** | **33.73** | **43.44** | 47.77 | **56.01** |
> | KVQuant-2b+1% sparse | 2.32 | **10.31** | **28.06** | 35.64 | 42.43 | **46.39** |
> | CQ-4c9b | 2.26 | **10.31** | 27.76 | **35.91** | **44.51** | 45.75 |
> | KVQuant-2b | 2.00 | 2.27 | 9.85 | 12.55 | 20.18 | 19.94 |
> | CQ-4c8b | 2.00 | **8.04** | **25.67** | **30.89** | **45.4** | **41.94** |
> | KVQuant-1b+1% sparse | 1.32 | 2.27 | 10.75 | 14.09 | 20.77 | 19.94 |
> | CQ-8c10b | 1.27 | **2.35** | **13.13** | **21.81** | **28.19** | **26.98** |
> | KVQuant-1b | 1.00 | 0.68 | 0 | 0 | 0 | 0 |
> | CQ-8c8b | 1.00 | **1.74** | **5.37** | **11.39** | **20.77** | **16.72** |
>
> **[W2] Incomplete Explanation of Channel Correlation: The paper doesn't fully explain the existence and variability of channel correlation across different layers, which may raise questions about the method's general applicability.**
>
> - The amount of correlation between channels does vary across layers. In the table below, we present the mean absolute correlation (MAC), excluding the diagonals, of different layers of LLaMA-7b on 262K tokens of WikiText-2.
> - Although the key correlation of the first few layers are higher than the later layers, the key/value correlation of any layer is never very close to zero, meaning channel coupling can be effective for any layer.
> - Correlation as a metric only captures the linear dependency between a pair of channels. In practice, we couple up to 8 channels together to leverage the higher order dependency between multiple channels.
> - We thank the reviewer for the careful reading of our paper. We will include a discussion on the existence and variability of channel correlation across different layers in the final paper.
>
> | Mean Absolute Correlation |  |  |  |  |  |  |  |  |
> |---|---|---|---|---|---|---|---|---|
> | Layer | 1 | 2 | 3 | 4 | 5 | 6 | 7 | 8 |
> | Key | 0.407 | 0.212 | 0.193 | 0.178 | 0.114 | 0.113 | 0.115 | 0.122 |
> | Value | 0.071 | 0.084 | 0.061 | 0.073 | 0.055 | 0.056 | 0.056 | 0.057 |
> | Layer | 9 | 10 | 11 | 12 | 13 | 14 | 15 | 16 |
> | Key | 0.131 | 0.138 | 0.090 | 0.098 | 0.094 | 0.141 | 0.109 | 0.114 |
> | Value | 0.061 | 0.067 | 0.047 | 0.042 | 0.065 | 0.065 | 0.062 | 0.035 |
> | Layer | 17 | 18 | 19 | 20 | 21 | 22 | 23 | 24 |
> | Key | 0.136 | 0.111 | 0.091 | 0.101 | 0.102 | 0.156 | 0.071 | 0.094 |
> | Value | 0.039 | 0.038 | 0.052 | 0.070 | 0.031 | 0.038 | 0.061 | 0.074 |
> | Layer | 25 | 26 | 27 | 28 | 29 | 30 | 31 | 32 |
> | Key | 0.107 | 0.069 | 0.057 | 0.103 | 0.095 | 0.097 | 0.100 | 0.090 |
> | Value | 0.027 | 0.044 | 0.038 | 0.072 | 0.070 | 0.035 | 0.105 | 0.090 |
>
> **[Q1] Quantization Setting: Is the KV Cache quantized during the prefill stage, or only during the decoding stage (after full-precision prefill computation), similar to the KIVI approach?**
>
> - For the majority of the experiments presented in the paper, including Table 1,2,4,5,6, all tokens are quantized in both the prefill and the decoding stage. We have specified this on line 247 of the paper.
> - For experiments with sliding window full-precision cache, including Table 3 and parts of Figure 1, we keep a constant number of the most recent tokens in full precision and quantize the rest of the tokens, and this holds true with respect to any token in the prefill and the decoding stage. This is slightly different from the KIVI implementation, which computes prefill in full precision and adopts a sliding window approach during decoding. We adopt this approach since perplexity testing and some tasks are log-likelihood-based and do not have a decoding stage.
>
> **[Q2] Channel Grouping Strategy: Would coupling non-adjacent channels with high mutual information potentially bring more benefits? --- Yes, very likely!**
>
> - Coupling channels with high mutual information will highly likely bring benefits. As shown in Figure 6 and 7, certain pairs of non-adjacent channels have higher correlations. By coupling them into the same channel group, we further reduce the joint entropy of channel groups, leading to better quantization accuracy.
> - We thank the reviewer for pointing us in this direction, and leave this for future work due to the difficulty of system implementation and optimizations.
>
> **[Q3] CUDA Kernel Implementation Questions**
>
> - Yes, the efficiency may be improved by adjusting the thread block assignments and reduction strategies.
> - First, it is important to note that the centroid size of each channel group is greater than 256 bytes. For example, for CQ-8c8b, the centroid size of each channel group is (num_centroids x num_channels x 2 bytes) $2^8 \times 8 \times 2=4096$ bytes. Assuming 100KB of shared memory per thread block, we can fit at most 25 groups of centroids into shared memory.
> - Fitting more channel groups into the same thread block reduces the number of concurrent writes to the HBM, hence speeding up the computation.
> - We thank the reviewer for this valuable suggestion. We will incorporate this kernel improvement into our implementation.

---

> > ### Comment · Reviewer_rmbd · 2024-08-10
> >
> > Thank you to the author for the detailed reply, which addressed most of my concerns. I am generally satisfied with the response.
> > I appreciate your detailed explanation of the CUDA kernel implementation. Previously, I had some misunderstandings regarding centroid storage, which have now been clarified. However, inefficiency remains an issue. From the latency measurements of CQ compared to KIVI in your rebuttal, it appears that the latency actually increased when reducing from 2 bits to 1 bit. This suggests that the overhead of lookup table might outweigh the benefits of reducing memory transfers..
> > Taking everything into consideration, I have decided to maintain my score of 6 points.

---

### Official Review · Reviewer_dwgV · 2024-07-12

**Soundness:** 4
**Presentation:** 4
**Contribution:** 3
**Rating:** 7
**Confidence:** 4

**Summary:**

This paper identifies a significant interdependency among distinct channels of key/value activation tensors in Transformer models. By quantizing multiple key/value channels together using joint entropy, the authors achieve high inference throughput while maintaining model quality. In extreme cases, the KV cache can be quantized down to 1 bit.

**Strengths:**

* The observation regarding the interdependency of KV channels is actually interesting and insightful.
* The proposed joint entropy-based quantization is intuitive and effective.
* The presentation is easy to follow and supported by comprehensive experiments.

**Weaknesses:**

* The improvement in inference throughput over the fp16 version is primarily observed with large batch sizes due to the high (de)quantization overhead.

**Questions:**

This paper is an enjoyable read, and I believe grouping K/V channels is an effective way to reduce memory storage burden and improves inference performance. Nevertheless, I have some questions listed below:

1. This work primarily focuses on quantizing the KV cache. Can it be combined with other quantization methods such as weight/activation quantization like SmoothQuant and AWQ? The authors mention that these methods are orthogonal to this work; could you elaborate on how they might be incorporated together?
2. "We employ the 'binning' trick [17] to estimate entropy. We first observe an empirical distribution of key and value channels by saving the KV cache on a dataset". Does this mean that the estimation needs to be done for each dataset and cannot be reused? Which dataset was used for the centroid learning in Table 8?
3. Section 3.3 seems to be a standard practice for fusing (de)quantization operations with other operations. The lookup table also does not seem efficient. Is it possible to conduct an ablation study or profiling to understand the overhead of centroid loading? Can these centroids be reused instead of being frequently loaded from global memory?
4. The experiment in Section 4.4 with large batch sizes is more relevant for online serving scenarios. It would be beneficial to integrate your method with [vllm](https://blog.vllm.ai/2023/06/20/vllm.html) or other serving frameworks and use real traces to evaluate performance. From Figure 4, it seems all CQ methods perform worse than the fp16 version with small batch sizes. Can you explain why? Is it due to the overhead of (de)quantization?
5. "This sliding window of full-precision cache only introduces a small constant memory overhead for each sequence." What about the latency overhead? How does the sliding window affect inference latency?
6. Based on Figure 1, it seems that only by combining the sliding window and CQ can a similar perplexity to the fp16 version be achieved. How do you determine the best sliding window size to achieve the optimal tradeoff? Also, what happens if the number of coupled channels exceeds 8?

**Limitations:**

Refer to the Weaknesses and Questions sections. Additionally, here are some minor issues:

* It would be useful to include latency experiments in Section 4.4 in addition to throughput, which can help readers better understand the benefits of the proposed method.
* It would be more helpful to show the actual memory usage in GB instead of the parameter count in Table 8.

---

> ### Author Rebuttal · Authors · 2024-08-07
>
> We sincerely thank the reviewer for the support of our paper and the insightful suggestions. We address the reviewer's concerns as follows.
>
> **[W1, L1] Include latency experiments.**
>
> - In the table below, we present additional latency measurements of CQ with comparison to FP16 cache and KIVI. We use Llama-2-7b with a batch size of 1 and a prompt of 2000 tokens to decode 100 tokens.
> - We observe that CQ achieves comparable efficiency with KIVI. We plan to further optimize CQ at system level to reduce latency and improve throughput.
>
> |  | Full-precision Sliding Window Length | Prefill Time (s) | Decoding Time (s) |
> |---|---|---|---|
> | FP16 | - | 0.853 | 0.0559 +/- 0.0044 |
> | KIVI-4b | 32 | 1.483 | 0.0693 +/- 0.0212 |
> | KIVI-2b | 32 | 1.291 | 0.0684 +/- 0.0213 |
> | CQ-2c8b | 0 | 1.820 | 0.0695 +/- 0.0056 |
> | CQ-2c8b | 32 | 1.926 | 0.0701 +/- 0.0057 |
> | CQ-4c8b | 0 | 1.684 | 0.0704 +/- 0.0056 |
> | CQ-4c8b | 32 | 1.790 | 0.0706 +/- 0.0058 |
> | CQ-8c8b | 0 | 1.726 | 0.0670 +/- 0.0070 |
> | CQ-8c8b | 32 | 1.857 | 0.0799 +/- 0.0066 |
>
> **[Q1] Combining with other weight/activation quantization methods.**
>
> - Our KV cache quantization method can be combined with other weight/activation quantization methods. In the standard CQ calibration process, the centroids are learned using the KV cache of the full-precision model based on a calibration dataset.
> - To combine CQ with other weight/activation quantization methods, we use the KV cache produced by the quantized model for centroid learning to minimize the distortions introduced by CQ.
>
> **[Q2] Does this mean that the estimation needs to be done for each dataset and cannot be reused? Which dataset was used for the centroid learning in Table 8? --- No, centroid learning only needs to be done once.**
>
> - Centroids of CQ are learned once during the calibration phase, and can be used for different downstream tasks.
> - For Table 8, the centroids of CQ are learned on a set of 16 sequences from WikiText-2, each with 2048 tokens. We have specified this on line 243 in the paper. The centroids are learned only once and used for all perplexity and accuracy experiments in the paper.
> - We present additional experiments below on the accuracy of CQ with different calibration datasets. We use 16 sequences of 2048 tokens from WikiText-2 and C4 as calibration set and evaluate CQ on 4 downstream tasks.
> - Despite using different calibration datasets, CQ performs similarly in various downstream tasks.
>
> |  | Calibration Dataset | WinoGrande | PIQA | Arc-C | GSM8K CoT |
> |---|---|---|---|---|---|
> | CQ-2c8b | WikiText-2 | 68.27 | 77.91 | 43.34 | 14.71 |
> |  | C4 | 68.35 | 77.86 | 43.16 | 14.71 |
> | CQ-4c8b | WikiText-2 | 66.45 | 76.12 | 39.93 | 8.04 |
> |  | C4 | 66.22 | 76.61 | 39.93 | 8.34 |
> | CQ-8c8b | WikiText-2 | 55.01 | 71.22 | 30.2 | 1.74 |
> |  | C4 | 56.27 | 71.55 | 30.52 | 1.9 |
>
> **[Q3] Ablation study on the overhead of centroid loading.**
>
> - We perform an additional ablation study to understand the overhead of centroid loading. We profile the KQ multiplication kernel for CQ-2c8b with a single query and 4K or 16K keys using Llama-2-7b hidden dimensions. We enable and disable loading centroids from global memory to shared memory. The results shown in the table below are average over 1000 kernel runs.
> - As shown in the table, centroid loading does not significantly contribute to the overall latency. The latency primarily comes from reading quantized KV cache and queries from global memory, and writing the results to global memory. Hence reusing the centroids will not significantly reduce the latency.
>
> |  | Sequence Length = 4000 | Sequence Length = 16000 |
> |---|---|---|
> | Centroid Loading | 214.549 us +/- 12.353 | 556.184 us +/- 10.970 |
> | No Centroid Loading | 207.896 us +/- 11.544 | 548.111 us +/- 7.917 |
>
> **[Q4] Integration with vllm or other serving frameworks. From Figure 4, it seems all CQ methods perform worse than the fp16 version with small batch sizes. Can you explain why? Is it due to the overhead of (de)quantization?**
>
> - We anticipate compatibility between our proposed CQ and PagedAttention in vLLM, and believe their integration is a promising avenue for future exploration. Given the implementation difficulty, we leave this to future investigations.
> - CQ has higher latency than FP16 in small batch sizes due to the overhead of (de)quantization. As suggested by Reviewer rmbd, the efficiency of our kernel can be further optimized by loading centroids of more channel groups at once into shared memory, and adjusting the thread block assignment and reduction strategies. We will incorporate these improvements after the rebuttal period.
>
> **[Q5] How does the sliding window affect inference latency?**
>
> - Please see the first table for a latency comparison between sliding window and no sliding window. Sliding window full-precision cache does not significantly contribute to the overall latency in the prefill or the decoding stage.
>
>
> **[Q6] How do you determine the best sliding window size to achieve the optimal tradeoff? Also, what happens if the number of coupled channels exceeds 8?**
>
> - The sliding window size needs to be larger for lower bit widths to compensate for the precision loss. For CQ-2c8b (4-bit quantization), a small window size of 16 tokens or no window may suffice, but for CQ-8c8b (1-bit quantization), a larger window of 128 tokens may be necessary for preserving quality.
> - The number of coupled channels can exceed 8. We present additional experimental results below using LLaMA-7b with CQ-16c12b (12 bits per 16 coupled channels, averaging 0.81 bits per activation).
>
> |  | BPA | WikiText-2 PPL | C4 PPL |
> |---|---|---|---|
> | CQ-16c12b | 0.81 bits | 8.71 | 14.40 |
>
> **[L2] It would be more helpful to show the actual memory usage in GB instead of the parameter count in Table 8.**
>
> - We will include the actual memory usage in the final paper. The memory overhead of centroids can be calculated as (parameter count x 2) bytes.

---

> > ### Comment · Reviewer_dwgV · 2024-08-10
> >
> > Thank you for your detailed response. Most of my concerns have been addressed. Based on the latency results, there remains a large gap between CQ and the fp16 version (especially for the prefill stage). If compared with the quantized fp8 or int8 implementation, I think the performance gap will be larger, so I hope the authors can further enhance efficiency to make this quantization technique more practical. For now, I will maintain my current score.

---

### Official Review · Reviewer_qfND · 2024-07-12

**Soundness:** 3
**Presentation:** 3
**Contribution:** 2
**Rating:** 6
**Confidence:** 4

**Summary:**

The paper explores the idea of compressing the KV cache in Transformer models through quantization; specifically, the authors propose Coupled Quantization, which quantizes multiple KV channels together in order to exploit their interdependency. The gains are guaranteed by the fact that the joint entropy is smaller than or equal to the sum of marginal entropies of the channels. The channels are coupled in continuous groups. The experimental results show that CQ maintains the model quality and improves the inference throughput, even with high compression rates.

**Strengths:**

The paper uses known information theoretic techniques and applies them to KV cache quantization.
The method is well founded in information theory.
The experimental evaluation is comprehensive, showing significant improvements in inference throughput.
The method is relevant for a real bottleneck in deploying LLMs, which is the GPU memory usage due to KV caching.

**Weaknesses:**

While the experiments are extensive, they are focused on the Llama family of models.
The novelty of the paper is limited.

**Questions:**

Does CQ affect the training process or is it purely an inference-time optimization?

Do you envision a way of using non-contiguous channels in the coupling, perhaps with some limited search of finding groups of channels with higher interdependency?

Can you clarify the meaning of bold and underline in Tables?

**Limitations:**

Yes.

---

> ### Author Rebuttal · Authors · 2024-08-07
>
> We sincerely thank the reviewer for their careful review of our paper and the insightful suggestions. We address your concerns as follows.
>
> **[W1] Focused on the Llama family of models. -- We have added Mistral model results!**
>
> - We would like to draw the reviewer's attention to the results on Mistral in Tables 1, 2, and 6.
>
> **[W2] The novelty of the paper is limited.**
>
> Although novelty is a multifaceted concept in research, we believe it can be viewed roughly from two aspects: ***empirical novelty***, which involves the discovery of properties unknown to the community (e.g. the lottery ticket hypothesis [1]), and ***technical novelty***, which refers to the development of new solutions (e.g. Attention [2]). We argue that our work exhibits both forms of novelty, as detailed below.
>
> - **Empirically Novel Observation that KV Channels are Highly Interdependent:** To the best of our knowledge, we are the first to highlight the phenomenon that channels of KV cache exhibit high amounts of mutual information or correlation. This discovery opens up new avenues for KV cache compression.
> - **Technically Novel Approach that Enables Extreme KV Cache Compression:** Based on our novel observation and concepts from Information Theory, we propose a new approach for KV cache quantization by coupling multiple KV channels. Existing approaches such as KVQuant and KIVI quantize KV cache channel-wise or token-wise independently, which cannot take advantage of the high mutual information shared between channels. By exploiting the interdependency among channels, we enable KV cache compression rates previously difficult or impossible to achieve, i.e., 1-bit quantization of KV cache.
>
>
> **[Q1] Does CQ affect the training process or is it purely an inference-time optimization? --- It does not!**
>
> - CQ is a post-training quantization (PTQ) approach that does not affect the training process of the models. CQ can improve the inference efficiency by saving memory and increasing batch size.
>
> **[Q2] Do you envision a way of using non-contiguous channels in the coupling, perhaps with some limited search of finding groups of channels with higher interdependency? --- Yes!**
>
> - Coupling non-contiguous, highly interdependent channels will likely further improve the quantization accuracy. One potential way of achieving that is to place channels with the highest correlation into the same channel group through a greedy search. We leave this for future work due to the difficulty in system implementation and optimizations.
> - We thank the reviewer for pointing us in this direction.
>
> **[Q3] Can you clarify the meaning of bold and underline in Tables? --- Sure!**
>
> - The bolded number is the best perplexity/accuracy achieved under the same bit-width, while the underlined number is the second best. We will clarify this in the final paper.
>
> **References**
>
> [1] Frankle, Jonathan, and Michael Carbin. "The lottery ticket hypothesis: Finding sparse, trainable neural networks." arXiv preprint arXiv:1803.03635 (2018).
>
> [2] Vaswani, Ashish, et al. "Attention is all you need." Advances in neural information processing systems 30 (2017).

---

### Official Review · Reviewer_KwWk · 2024-07-13

**Soundness:** 2
**Presentation:** 1
**Contribution:** 2
**Rating:** 6
**Confidence:** 3

**Summary:**

The authors have addressed the KV-cache compression problem by providing a finer quantization level. The KV-cache can pose a significant barrier to the inference of most autoregressive language models, a challenge that has been well studied in recent publications at ICML and NeurIPS. This paper introduces a novel approach by coupling multiple key/value channels together for quantization, exploiting their interdependence to encode the activations in a more information-efficient manner.

**Strengths:**

- The method is novel and demonstrates comparable accuracy to KVQuant, one of the pivotal approaches in this field.
- It includes a substantial number of experiments to validate accuracy.
- The quantization method introduced here is novel compared to other approaches. Additionally, the implementation in PyTorch represents a significant contribution.

**Weaknesses:**

- The code is not available. For research on KV cache, it is important to have the code available.

- The section describing the random variables and entropy, specifically line 120, does not explicitly describe the random variables in mathematical notation. This should be revised for clarity. I would like to see this section more polished.

- I believe the LLAMA3 model was available before the NeurIPS submission. Since that time, the authors may have extended their findings to these models. I would like to see the performance of your model in that setting.

- I want to see how the runtime of your method compares to other methods. Recent works, like **QJL**, include good plots for token-wise generation time or end-to-end timing. Since you compare with KVQuant, it is also good to compare with **KIVI**, as it is one of the best methods. I recommend comparing with QJL and **KIVI**, and plotting the runtime alongside these methods.

- I highly recommend the authors run their code on longer context datasets. LongBench could be a great example to evaluate its performance compared to other methods. I would suggest that perplexity is not the best metric for comparison.

Relevant papers: [KIVI: A Tuning-Free Asymmetric 2bit Quantization for KV Cache], [QJL: 1-Bit Quantized JL Transform for KV Cache Quantization with Zero Overhead]

**If you address the concerns regarding the experiments and provide a broader comparison to the other methods, I would increase my score.**

**Questions:**

- There is additional overhead regarding storing the centroid for each coupled key/value pair, making it difficult to track. I would like you to mention this overhead and explain how you set those values.

**Limitations:**

- There is additional overhead regarding storing the centroid for each coupled key/value pair, which can complicate tracking and management. It would be beneficial to address this overhead and provide details on how these values are set.

---

> ### Author Rebuttal · Authors · 2024-08-07
>
> We express our sincere gratitude to the reviewer for their thoughtful comments and suggestions. We address the reviewer's concerns as follows.
>
> **[W1] The code is not available.**
>
> - We will open source our code during the camera-ready phase. To provide additional context, we have included expanded experimental results and implementation profiling details below.
>
> **[W2] Section 3 should be revised for clarity.**
>
> - We will revise and polish section 3 to describe the random variables and entropy in mathematical notations. We appreciate the reviewer's careful reading of our paper and their valuable feedback.
>
> **[W3] Extension to LLAMA3.**
>
> - We present additional experimental results comparing CQ and KVQuant on LLaMA-3-8b in the table below. CQ mostly outperforms KVQuant, especially in lower bit widths. We would like to gently remind that LLaMA-3 was released only one month before the NeurIPS deadline.
>
> |  | BPA | WikiText-2 PPL | WinoGrande | PIQA | Arc-C |
> |---|---|---|---|---|---|
> | FP16 | 16 | 5.54 | 72.69 | 79.71 | 50.51 |
> | KVQuant-4b | 4 | 5.66 | 72.77 | **79.98** | 47.44 |
> | CQ-2c8b | 4 | **5.58** | **73.16** | 78.84 | **49.83** |
> | KVQuant-2b | 2 | 18.96 | 56.27 | 63.49 | 24.4 |
> | CQ-4c8b | 2 | **6.09** | **69.22** | **78.62** | **44.03** |
> | KVQuant-1b | 1 | 22238.91 | 50.04 | 53.05 | 22.35 |
> | CQ-8c8b | 1 | **9.56** | **56.04** | **72.58** | **32.51** |
>
> **[W4] Comparison with QJL and KIVI.**
>
> - We tried our best to compare with QJL in the limited timeframe of the rebuttal. However, we ran into some issues due to incompatibilities of the QJL codebase with our hardware. Specifically, we ran into the following error: `File ".../QJL/models/llama3_utils_qjl.py", line 138, in build_sketch
>     self.key_states_norm = torch.norm(key_states, dim=-1)
> RuntimeError: CUDA error: limit is not supported on this architecture`. This could be caused by incompatibilities of our V100 GPUs with the code. We will try to resolve this issue during the discussion period. We would like to kindly remind that QJL was first published online in June, which is after the NeurIPS deadline.
> - We present additional experiments in the table below comparing CQ with KIVI on accuracy of LongBench with Llama-2-7b. For KIVI, we use 2-bit quantization, a full-precision sliding window (residual length) of 32 tokens, and a group size of 32. For CQ, we use 2-bit quantization and a sliding window size of 32. CQ mostly outperforms KIVI across different tasks.
>
> |  | Sliding Window Size | Qasper | QMSum | MultiNews | TREC | TriviaQA | SAMSum | LCC | RepoBench-P |
> |---|---|---|---|---|---|---|---|---|---|
> | FP16 | - | 9.52 | 21.28 | 3.51 | 66.00 | 87.72 | 41.69 | 66.66 | 59.82 |
> | KIVI-2 | 32 | 9.26 | 20.53 | 0.97 | **66.00** | 87.42 | **42.61** | 66.22 | 59.67 |
> | CQ-4c8b | 32 | **9.58** | **20.87** | **1.93** | **66.00** | **87.72** | 41.13 | **66.57** | **59.75** |
>
> - In the table below, we present additional latency measurements of CQ with comparison to KIVI and FP16 cache. We use Llama-2-7b with a batch size of 1 and a prompt of 2000 tokens to decode 100 tokens.
>
> |  | Full-precision Sliding Window Length | Prefill Time (s) | Decoding Time (s) |
> |---|---|---|---|
> | FP16 | - | 0.853 | 0.0559 +/- 0.0044 |
> | KIVI-4b | 32 | 1.483 | 0.0693 +/- 0.0212 |
> | KIVI-2b | 32 | 1.291 | 0.0684 +/- 0.0213 |
> | CQ-2c8b | 0 | 1.820 | 0.0695 +/- 0.0056 |
> | CQ-2c8b | 32 | 1.926 | 0.0701 +/- 0.0057 |
> | CQ-4c8b | 0 | 1.684 | 0.0704 +/- 0.0056 |
> | CQ-4c8b | 32 | 1.790 | 0.0706 +/- 0.0058 |
> | CQ-8c8b | 0 | 1.726 | 0.0670 +/- 0.0070 |
> | CQ-8c8b | 32 | 1.857 | 0.0799 +/- 0.0066 |
>
> **[W5] Longer context datasets such as LongBench.**
>
> - Please see the table above for CQ's results on LongBench with comparison to KIVI.
> - We present additional experiments with longer context datasets comparing CQ and KVQuant using Llama-2-7b on GSM8K with chain-of-thought (CoT) and MMLU with CoT Fewshot (gsm8k_cot, mmlu_flan_cot_fewshot_humanities, mmlu_flan_cot_fewshot_stem, mmlu_flan_cot_fewshot_social_sciences, mmlu_flan_cot_fewshot_other from lm-evaluation-harness). In long-context settings, CQ mostly outperforms or performs similarly to KVQuant under the same bit width.
>
> |  | BPA | GSM8K CoT | MMLU (STEM) CoT Fewshot | MMLU (Humanities) CoT Fewshot | MMLU (Social Sciences) CoT Fewshot | MMLU (Other) CoT Fewshot |
> |---|---|---|---|---|---|---|
> | KVQuant-4b+1% sparse | 4.32 | 14.33 | 31.04 | 41.12 | **48.37** | 55.43 |
> | CQ-2c8b | 4.00 | **14.71** | **33.73** | **43.44** | 47.77 | **56.01** |
> | KVQuant-2b+1% sparse | 2.32 | **10.31** | **28.06** | 35.64 | 42.43 | **46.39** |
> | CQ-4c9b | 2.26 | **10.31** | 27.76 | **35.91** | **44.51** | 45.75 |
> | KVQuant-2b | 2.00 | 2.27 | 9.85 | 12.55 | 20.18 | 19.94 |
> | CQ-4c8b | 2.00 | **8.04** | **25.67** | **30.89** | **45.4** | **41.94** |
> | KVQuant-1b+1% sparse | 1.32 | 2.27 | 10.75 | 14.09 | 20.77 | 19.94 |
> | CQ-8c10b | 1.27 | **2.35** | **13.13** | **21.81** | **28.19** | **26.98** |
> | KVQuant-1b | 1.00 | 0.68 | 0 | 0 | 0 | 0 |
> | CQ-8c8b | 1.00 | **1.74** | **5.37** | **11.39** | **20.77** | **16.72** |
>
> **[Q1, L1] Overhead regarding storing the centroids. I would like you to mention this overhead and explain how you set those values.**
>
> - We kindly refer the reviewer to Section A.4 for a discussion on the overhead of storing and learning the centroids. We have reported the number of centroid parameters and the learning time for each CQ configuration and model, and described how the overhead is calculated in detail.
> - CQ only has two hyperparameters: the number of channels and the number of bits in a code. These parameters are explicitly reported in our experimental results. There is no need of extensive hyperparameter tuning for our method.

---

> ### Author Response · Authors · 2024-08-12
> **QJL Experiment Results & Follow-up**
>
> We thank the reviewer again for carefully reviewing our paper and providing constructive feedback. We have conducted additional experiments using the official QJL codebase (https://github.com/amirzandieh/QJL) with an A100 40GB GPU. We evaluate QJL and CQ with Llama-2-7b on LongBench.
> For QJL, we used a sliding window of size 32 and a group size of 32  (`buffer_size=32,group_size=32`), and set other hyper-parameters following the codebase (`key_quantization_bits=256,key_quantization_bits_initial_layers=512,initial_layers_count=15,outlier_count_general=8,outlier_count_initial_layers=8,value_quantization_bits=2`). For CQ, we use the 4c8b (2-bit) configuration and a sliding window of size 32. The results are presented in the table below.
>
> |  | Bit Width | Qasper | TREC | SAMSum | TriviaQA |
> |---|---|---|---|---|---|
> | FP16 | 16 | 9.52 | 66.00 | 41.69 | 87.72 |
> | QJL | 3.00 | 5.98 | 15.00 | 14.84 | Error |
> | CQ-4c8b | 2.00 | **9.58** | **66.00** | **41.13** | **87.72** |
>
> We also encountered some challenges when attempting to conduct additional experiments.
>
> 1. For QJL on TriviaQA, we ran into the following error: `File ".../QJL/models/llama2_utils_qjl.py", line 143, in _update_outliers
>     self.outlier_indices = torch.cat([self.outlier_indices, outlier_indices], dim=2).contiguous()
> TypeError: expected Tensor as element 0 in argument 0, but got NoneType`.
>
> 2. We tried to directly compare CQ with QJL by following the experimental settings (longchat-7b-v1.5-32k on LongBench) in Table 1 of the QJL paper. However, we ran into out-of-memory issues with CQ-4c8b due to GPU memory constraints (Nvidia A100 40G).
>
> Given the time constraint of the discussion period, we have made our best effort to provide a fair comparison between CQ and QJL. We are open to further investigation and would welcome specific suggestions from the reviewer on how to resolve the issues above. We are also happy to provide additional clarification on any follow-up questions. We respectfully request that the reviewer reconsider our paper in light of these responses.

---

> ### Author Response · Authors · 2024-08-14
> **Thank you to Reviewer**
>
> Dear Reviewer KwWk,
>
> We would like to express our sincere gratitude for your time and effort in reviewing our paper. Your feedback has been invaluable to us.
>
> As the discussion period is drawing to a close, we kindly request that you review our previous responses to your review. If you have any additional questions or concerns, we would be happy to address them promptly.
>
> Thank you again for your valuable contributions.

---

> > ### Comment · Reviewer_KwWk · 2024-08-14
> > **Score Increased to 6**
> >
> > I'm glad the answer has addressed my questions. Extending the experiments and comparing them to the baselines, including LongBench and LLaMA3, is great! I'll be more than happy to increase my score to 6.
> >
> > I recommend adding all of these results to the main paper.

---

> > > ### Author Response · Authors · 2024-08-14
> > > **Thank You to Reviewer**
> > >
> > > We sincerely thank the reviewer for their thoughtful comments and suggestions!

---

### Official Review · Reviewer_p3Eo · 2024-07-30

**Soundness:** 3
**Presentation:** 3
**Contribution:** 2
**Rating:** 5
**Confidence:** 4

**Summary:**

LLM inference typically involves Key-Value (KV) caching to avoid recomputation. However, the KV cache size grows with batch and context length, imposing bottlenecks in memory footprint and inference speed. Quantization can be employed to reduce the size of the KV cache. Instead of channel-wise independent quantization, this work proposes Coupled Quantization to quantize the KV cache by grouping the channels and performing vector quantization by learning a joint codebook with a calibration corpus. This enables additional compression as the entropy of each channel is distributed over the code bits.

**Strengths:**

- With the proposed method, more aggressive compression of the KV cache is enabled.
- System support is provided.

**Weaknesses:**

- Table 2 shows that CQ only outperforms the baselines in the 1bit regime, which already suffers great accuracy loss. For relatively “safer” bitwidths, it is on-par or sub-par to baselines.
- As far as I know, the common sense reasoning tasks utilized in the experiments(Windogrande, PIQA, ARC, etc.) are usually multiple choice and do not require long generation (also, they are relatively easy tasks). It could be hard to verify the effectiveness of the proposed method for long generation setting(for instance, experiments on benchmarks such as GSM8k with chain-of-thought, which is a relatively hard task), where the incoming KVs are also quantized.
- Baselines are absent in Table 3.
- Testing the proposed method on long context benchmarks or multi-hop retrieval tasks, such as the RULER[1] dataset will improve the quality of the paper.
- The proposed method involves offline learning of centroids using a calibration dataset. The quality of the centroids may not hold under distribution shifts.

Reference:

[1] Hsieh et al, “RULER: What's the Real Context Size of Your Long-Context Language Models?”, arXiv 2024.

**Questions:**

- How are shared memory bank conflicts handled during centroid lookup?

**Limitations:**

The authors have included a separate Limitations & Broader Impacts section in the manuscript.

---

> ### Author Rebuttal · Authors · 2024-08-07
>
> We sincerely thank the reviewer for their careful consideration of our work and for providing valuable feedback. We have addressed their comments in detail below.
>
> **[W1] Table 2 shows that CQ only outperforms the baselines in the 1bit regime, which already suffers great accuracy loss.**
>
> - It is essential to consider the bits per activation metric when comparing quantization methods in Table 2. Under identical whole-number bit widths (4.00 bits, 2.00 bits, and 1.00 bit), CQ consistently surpasses KVQuant in average accuracy, particularly in the 2-bit and 1-bit regimes where the improvement exceeds 10%. When considering similar bit widths, CQ outperforms or matches the performance of sparsity-based KVQuant.
> - We highlight that, in Table 2, CQ-8c8b (with a bit width of 1) outperforms KVQuant-2b (with a bit width of 2) in average accuracy despite using half the memory, demonstrating the efficacy of our proposed approach.
> - Additionally, it's important to note that KVQuant+1% sparse is a sparsity-based method that relies on storing outlier activations in a sparse matrix. This approach can potentially introduce latency overhead and deployment complexities. In contrast, our CQ method maintains a fully dense representation, avoiding these potential issues.
>
> **[W2] Evaluations with long generation.**
>
> - We present additional experimental results below with long-context datasets comparing CQ and KVQuant using Llama-2-7b on GSM8K with chain-of-thought (CoT) and MMLU with CoT Fewshot (gsm8k_cot, mmlu_flan_cot_fewshot_humanities, mmlu_flan_cot_fewshot_stem, mmlu_flan_cot_fewshot_social_sciences, mmlu_flan_cot_fewshot_other from lm-evaluation-harness). In long-context settings, CQ mostly outperforms or is comparable to KVQuant under the same bit width.
>
> |  | BPA | GSM8K CoT | MMLU (STEM) CoT Fewshot | MMLU (Humanities) CoT Fewshot | MMLU (Social Sciences) CoT Fewshot | MMLU (Other) CoT Fewshot |
> |---|---|---|---|---|---|---|
> | KVQuant-4b+1% sparse | 4.32 | 14.33 | 31.04 | 41.12 | **48.37** | 55.43 |
> | CQ-2c8b | 4.00 | **14.71** | **33.73** | **43.44** | 47.77 | **56.01** |
> | KVQuant-2b+1% sparse | 2.32 | **10.31** | **28.06** | 35.64 | 42.43 | **46.39** |
> | CQ-4c9b | 2.26 | **10.31** | 27.76 | **35.91** | **44.51** | 45.75 |
> | KVQuant-2b | 2.00 | 2.27 | 9.85 | 12.55 | 20.18 | 19.94 |
> | CQ-4c8b | 2.00 | **8.04** | **25.67** | **30.89** | **45.4** | **41.94** |
> | KVQuant-1b+1% sparse | 1.32 | 2.27 | 10.75 | 14.09 | 20.77 | 19.94 |
> | CQ-8c10b | 1.27 | **2.35** | **13.13** | **21.81** | **28.19** | **26.98** |
> | KVQuant-1b | 1.00 | 0.68 | 0 | 0 | 0 | 0 |
> | CQ-8c8b | 1.00 | **1.74** | **5.37** | **11.39** | **20.77** | **16.72** |
>
> **[W3] Baselines are absent in Table 3.**
>
> - To add more baselines to Table 3, we present additional experiments on LongBench below comparing CQ and KIVI with sliding window full-precision cache using Llama-2-7b. For KIVI, we use 2-bit quantization, a full-precision sliding window (residual length) of 32 tokens, and a group size of 32. For CQ, we use 2-bit (4c8b) quantization and a sliding window size of 32. CQ mostly outperforms KIVI across different tasks.
>
> |  | Sliding Window Size | Qasper | QMSum | MultiNews | TREC | TriviaQA | SAMSum | LCC | RepoBench-P |
> |---|---|---|---|---|---|---|---|---|---|
> | FP16 | - | 9.52 | 21.28 | 3.51 | 66.00 | 87.72 | 41.69 | 66.66 | 59.82 |
> | KIVI-2b | 32 | 9.26 | 20.53 | 0.97 | **66.00** | 87.42 | **42.61** | 66.22 | 59.67 |
> | CQ-4c8b | 32 | **9.58** | **20.87** | **1.93** | **66.00** | **87.72** | 41.13 | **66.57** | **59.75** |
>
> **[W4] Testing the proposed method on long context benchmarks or multi-hop retrieval tasks.**
>
> - Please see the first table above for evaluations on long-context benchmarks including GSM8k with CoT and MMLU with CoT Fewshot.
> - We tried our best to evaluate our method on the RULER dataset in the limited timeframe of the rebuttal. However, we could not run RULER on our servers due to compatibility issues with docker. We will keep trying our method on RULER during the discussion period.
>
> **[W5] The quality of the centroids may not hold under distribution shifts.**
>
> - We have added an ablation study as follows to suggest that calibration on language modeling datasets provides transferable performance on downstream tasks. We use 16 sequences of 2048 tokens from WikiText-2 and C4 as the calibration set and evaluate CQ on 4 downstream tasks.
> - Despite using different calibration datasets, CQ performs similarly in various downstream tasks.
>
> |  | Calibration Dataset | WinoGrande | PIQA | Arc-C | GSM8K CoT |
> |---|---|---|---|---|---|
> | CQ-2c8b | WikiText-2 | 68.27 | 77.91 | 43.34 | 14.71 |
> |  | C4 | 68.35 | 77.86 | 43.16 | 14.71 |
> | CQ-4c8b | WikiText-2 | 66.45 | 76.12 | 39.93 | 8.04 |
> |  | C4 | 66.22 | 76.61 | 39.93 | 8.34 |
> | CQ-8c8b | WikiText-2 | 55.01 | 71.22 | 30.2 | 1.74 |
> |  | C4 | 56.27 | 71.55 | 30.52 | 1.9 |
>
> **[Q1] How are shared memory bank conflicts handled during centroid lookup?**
>
> - Due to a relatively high number of centroids (256 centroids for 8-bit codes and 1024 centroids for 10-bit codes), we did not notice shared memory bank conflicts to significantly impact system performance. We thank the reviewer for raising this insightful question and will continue to explore potential optimizations in this area.

---

> > ### Comment · Reviewer_p3Eo · 2024-08-14
> >
> > The accuracy of the proposed method is better compared to KVQuant on benchmark tasks. Although I think (saturated) latency comparisons with KVQuant or KIVI should be provided, Due to the discussion phase ending soon, I preemptively raise my score.

---

> ### Author Response · Authors · 2024-08-13
> **Additional Experiments & Follow-up**
>
> We appreciate the reviewer for carefully reviewing our paper and offering thoughtful feedback. We have conducted additional experiments on passkey retrieval, following the setup in [1], with Llama-2-7b at its maximum context length of 4096. The passkey retrieval task is similar to the needle retrieval task in RULER. We still have trouble running RULER due to compatibility issues with Docker, and we will continue to work on it. As shown in the table below, CQ consistently outperforms KVQuant at various bit widths on passkey retrieval. We are also happy to provide additional clarification on any follow-up questions. We respectfully request that the reviewer reconsider our paper in light of these responses.
>
> |  | Bit Width | Retrieval Success Rate |
> |---|---|---|
> | KVQuant-4b+1% sparse | 4.32 | **100%** |
> | KVQuant-4b | 4 | **100%** |
> | CQ-2c8b | 4 | **100%** |
> | KVQuant-2b+1% sparse | 2.32 | 94% |
> | CQ-4c9b | 2.26 | **98%** |
> | KVQuant-2b | 2 | 0% |
> | CQ-4c8b | 2 | **96%** |
> | KVQuant-1b+1% sparse | 1.32 | 2% |
> | CQ-8c10b | 1.27 | **78%** |
> | KVQuant-1b | 1 | 0% |
> | CQ-8c8b | 1 | **12%** |
>
> **References**
>
> [1] Zhu, Dawei, et al. "Pose: Efficient context window extension of llms via positional skip-wise training." arXiv preprint arXiv:2309.10400 (2023).

---

> > ### Comment · Reviewer_p3Eo · 2024-08-13
> >
> > Thank you for your response.
> >
> > Looking at the discourse of other reviewers, I have questions regarding the latency:
> >
> > I am confused as to why the latency is measured with batch size 1, while the manuscript presents a throughput increase with increasing batch sizes. For latency comparison, shouldn't token throughput or batched decoding latency comparison between CQ KIVI be measured in similar settings?

---

> > > ### Author Response · Authors · 2024-08-13
> > >
> > > We sincerely thank the reviewer for the response, and address your concerns as follows.
> > >
> > > - We presented the latency measurement with batch size 1, since Reviewer dwgV has specifically asked us for latency experiments in small batch sizes. We quote
> > > >From Figure 4, it seems all CQ methods perform worse than the fp16 version with small batch sizes. Can you explain why? Is it due to the overhead of (de)quantization?
> > > - Measurements with batch size 1 highlight the efficiency and latency aspects of our CUDA kernels.
> > > - We note that the latency comparison we presented between CQ and KIVI is a fair comparison with the same experimental settings: batch size of 1, equal token counts for prefill and decoding, and an identical sliding window size.
> > > - We agree with the reviewer that latency and throughput at different batch sizes are important for understanding the efficiency of our approach. Hence we will include additional latency and throughput measurements at different batch sizes and context lengths in the camera-ready version.

---

> > > > ### Comment · Reviewer_p3Eo · 2024-08-14
> > > >
> > > > Thank you for the response, but if measured for batch size 1, isn't the latency dominated by FFN?

---

### Author Rebuttal · Authors · 2024-08-07

We sincerely appreciate the reviewers' careful evaluation of our paper and their valuable feedback. In the following section, we address some common concerns raised by multiple reviewers. We are happy to provide further clarification during the discussion period.

**1. Evaluations with long-context benchmarks.**

- We present additional experimental results below with long-context datasets comparing CQ and KVQuant using Llama-2-7b, with all tokens quantized, on GSM8K with chain-of-thought (CoT) and MMLU with CoT Fewshot (gsm8k_cot, mmlu_flan_cot_fewshot_humanities, mmlu_flan_cot_fewshot_stem, mmlu_flan_cot_fewshot_social_sciences, mmlu_flan_cot_fewshot_other from lm-evaluation-harness). In long-context settings, CQ mostly outperforms or is comparable to KVQuant under the same bit width.

|  | BPA | GSM8K CoT | MMLU (STEM) CoT Fewshot | MMLU (Humanities) CoT Fewshot | MMLU (Social Sciences) CoT Fewshot | MMLU (Other) CoT Fewshot |
|---|---|---|---|---|---|---|
| KVQuant-4b+1% sparse | 4.32 | 14.33 | 31.04 | 41.12 | **48.37** | 55.43 |
| CQ-2c8b | 4.00 | **14.71** | **33.73** | **43.44** | 47.77 | **56.01** |
| KVQuant-2b+1% sparse | 2.32 | **10.31** | **28.06** | 35.64 | 42.43 | **46.39** |
| CQ-4c9b | 2.26 | **10.31** | 27.76 | **35.91** | **44.51** | 45.75 |
| KVQuant-2b | 2.00 | 2.27 | 9.85 | 12.55 | 20.18 | 19.94 |
| CQ-4c8b | 2.00 | **8.04** | **25.67** | **30.89** | **45.4** | **41.94** |
| KVQuant-1b+1% sparse | 1.32 | 2.27 | 10.75 | 14.09 | 20.77 | 19.94 |
| CQ-8c10b | 1.27 | **2.35** | **13.13** | **21.81** | **28.19** | **26.98** |
| KVQuant-1b | 1.00 | 0.68 | 0 | 0 | 0 | 0 |
| CQ-8c8b | 1.00 | **1.74** | **5.37** | **11.39** | **20.77** | **16.72** |

**2. Regarding Centroid Learning.**

- Centroids for CQ are learned once on a calibration dataset and can be used for different downstream tasks.
- We have added an ablation study as follows to suggest that calibration on language modeling datasets provides transferable performance on downstream tasks. We use 16 sequences of 2048 tokens from WikiText-2 and C4 as the calibration set and evaluate CQ on 4 downstream tasks.
- Despite using different calibration datasets, CQ performs similar in various downstream tasks.

|  | Calibration Dataset | WinoGrande | PIQA | Arc-C | GSM8K CoT |
|---|---|---|---|---|---|
| CQ-2c8b | WikiText-2 | 68.27 | 77.91 | 43.34 | 14.71 |
|  | C4 | 68.35 | 77.86 | 43.16 | 14.71 |
| CQ-4c8b | WikiText-2 | 66.45 | 76.12 | 39.93 | 8.04 |
|  | C4 | 66.22 | 76.61 | 39.93 | 8.34 |
| CQ-8c8b | WikiText-2 | 55.01 | 71.22 | 30.2 | 1.74 |
|  | C4 | 56.27 | 71.55 | 30.52 | 1.9 |

---

### Decision · Program_Chairs · 2024-09-25

**Decision:**

Accept (poster)

**Comment:**

The paper presents a novel approach to compressing the key-value (KV) cache in transformer models through a method called Coupled Quantization (CQ). This technique aims to enhance inference efficiency in large language models while maintaining model quality, addressing significant GPU memory constraints. The authors provide compelling experimental evidence that demonstrates substantial throughput improvements, which could be impactful for various applications in machine learning.
Overall, the reviewers provided a mix of support and concerns regarding the submission. While some reviewers agreed unanimously on the solid contributions of the paper, others expressed reservations about the novelty of the method and the limited scope of experiments primarily focused on certain transformer models, particularly those in the Llama family.
Despite the authors' thorough rebuttals, certain reviewer concerns remain unresolved, particularly regarding the generalizability of the findings across different model architectures and tasks. Reviewers noted the necessity for further validation in long-context settings and multi-hop retrieval tasks, as the current evaluations may not fully demonstrate CQ’s benefits. Additionally, the quality of the centroids under distribution shifts was questioned, signaling a need for more robust testing in diverse contexts.
It is vital that the authors address the reviewers' suggestions in the final version of the paper. These improvements, including expanded evaluation metrics and clarifications regarding the method's application, will significantly enhance the overall quality and impact of the research. I encourage the authors to implement these changes diligently to meet the expectations established through the review process.